# Elective nodal irradiation mitigates local and systemic immunity generated by combination radiation and immunotherapy in head and neck tumors

Laurel B. Darragh [1,2], Jacob Gadwa[1], Tiffany T. Pham[3], Benjamin Van Court[1], Brooke Neupert[1], Nicholas A. Olimpo[1], Khoa Nguyen[4], Diemmy Nguyen[1], Michael W. Knitz [1], Maureen Hoen[1], Sophia Corbo[1], Molishree Joshi[5], Yonghua Zhuang[6], Maria Amann[7], Xiao-Jing Wang [8,9], Steven Dow[10], Ross M. Kedl[1], Von Samedi[4], Mary-Keara Boss[10] & Sana D. Karam [1,2] ✉

In the setting of conventional radiation therapy, even when combined with immunotherapy, head and neck cancer often recurs locally and regionally. Elective nodal irradiation (ENI) is commonly employed to decrease regional recurrence. Given our developing understanding that immune cells are radio-sensitive, and that T cell priming occurs in the draining lymph nodes (DLNs), we hypothesize that radiation therapy directed at the primary tumor only will increase the effectiveness of immunotherapies. We find that ENI increases local, distant, and metastatic tumor growth. Multi-compartmental analysis of the primary/distant tumor, the DLNs, and the blood shows that ENI decreases the immune response systemically. Additionally, we find that ENI decreases antigen-specific T cells and epitope spreading. Treating the primary tumor with radiation and immunotherapy, however, fails to reduce regional recurrence, but this is reversed by either concurrent sentinel lymph node resection or irradiation. Our data support using lymphatic sparing radiation therapy for head and neck cancer.

Head and neck squamous cell carcinoma (HNSCC) is still primarily treated with radiation therapy (RT), chemotherapy, and surgery[1]. Therapeutic neck dissection and/or elective nodal irradiation (ENI) are utilized to minimize local and regional recurrence. Despite an aggressive treatment regimen, approximately 50% of patients with high-risk disease recur locally, regionally, or distantly by 3-years[2]. With the advent of immunotherapies, there was hope that patients with HNSCC would benefit from immune checkpoint inhibitors, but results of recent trials have dampened that hope[3,4]. Even in the setting of recurrent or metastatic disease, the benefit was non-existent or

[1]Department of Radiation Oncology, University of Colorado Denver at Anschutz Medical Campus, Aurora, CO, USA. [2]Department of Immunology and Microbiology, University of Colorado Denver at Anschutz Medical Campus, Aurora, CO, USA. [3]Department of Otolaryngology Head and Neck Surgery, University of Colorado Denver at Anschutz Medical Campus, Aurora, CO, USA. [4]Department of Pathology, University of Colorado Denver at Anschutz Medical Campus, Aurora, CO, USA. [5]Department of Pharmacology, University of Colorado Denver at Anschutz Medical campus, Aurora, CO, USA. [6]Department of Pediatrics, Cancer Center Biostatistics Core, University of Colorado Anschutz Medical campus, Aurora, CO, USA. [7]Roche Innovation Center Zurich, Roche Pharmaceutical Research and Early Development (pRED) Schlieren, Zurich, Switzerland. [8]Department of Pathology and Laboratory Medicine, University of California Davis, School of Medicine, Davis, USA. [9]Veterans Affairs Medical Center, VA Eastern Colorado Health Care System, Aurora, CO 80045, USA. [10]Department of Radiation Oncology, Colorado State University, Fort Collins, Colorado. Campus, Aurora, CO, USA. ✉e-mail: sana.karam@cuanschutz.edu

modest at best[5]. Why this is the case for a tumor subtype with relatively high tumor mutational burden (TMB) and moderate infiltration of immune cells in the tumor microenvironment (TME), remains an enigma for the field.

Improving response to immunotherapy requires a re-evaluation of its integration into standard-of-care treatment. Comprehensive nodal dissections and irradiation of involved lymph nodes are known to reduce the recurrence of regional and distant metastases. In the absence of sentinel lymph node mapping, however, it is often challenging for clinicians to evaluate the extent of nodal disease involvement. Neck dissections and elective nodal irradiation (ENI) are therefore commonly adopted strategies to decrease regional and distant metastasis[6]. Such targeting of tumor draining lymph nodes (DLNs), a site for T cell immune priming[7–11], could lead to a considerable decrease in the number of immune cells migrating to the TME for antigen-specific cell kill. Since immunotherapies are inherently reliant on the immune system, and the lymphatics, some have argued that immunotherapies may be more effective without comprehensive neck dissections or ENI[12–14].

In this work, we seek to determine if ENI or neck dissections are detrimental to the immune response generated by combining stereotactic body radiation (SBRT) with immunotherapy. To test this, we develop a preclinical radiation protocol where we are able to specifically target only the primary tumor in one group and the tumor plus the bilateral neck in a second group (ENI). We extend our findings of ENI in HNSCC to preclinical models of breast cancer and melanoma and develop a surgical model for a neck dissection and for sentinel lymph node resection. We test whether mice treated with SBRT that targets only the primary tumor have significantly better local, distant, and metastatic tumor control than mice that receive ENI or a neck dissection. We also investigate the immunological response to ENI in the blood, primary tumor, primary tumor DLNs, secondary tumors, and secondary tumor DLNs to provide perspective on how ENI affects the immune response systemically. In the context of combination radiation and immunotherapy, a major finding is that ENI increases local and distant failure by decreasing the systemic CD4 and CD8 effector T cells responsible for tumor control. Interestingly, mice that eradicate local and distant tumors after SBRT to the tumor only often recur regionally, but concurrent sentinel lymph node resection or sentinel lymph node irradiation is sufficient at mitigating regional spread while preserving the local and systemic immune response. Finally, specimen analysis from a recently completed clinical trial in canine HNSCC patients and a Phase I/Ib trial in human oral squamous cell carcinoma (NCT03635164), further validate our mouse findings that tumor only SBRT can generate a systemic immune response, particularly by mounting a CD8 effector T cell response. We believe our data demonstrate the importance of the DLNs in initial T cell priming and supports reducing ENI and comprehensive neck dissections in the clinical setting.

## Results

### ENI ablates the immune response to combined radiation and immunotherapy

To test if ENI dampens the anti-tumor immune response to SBRT combined with immunotherapy, we developed a radiation protocol that allowed us to target only primary gross tumor with or without bilateral lymph node irradiation (Supplementary Fig. 1A). To investigate if ENI decreased local and systemic immunity, we implanted LY2 HNSCC cells both into the buccal, representative of a primary orthotopic tumor, and into the flank, to represent a distant tumor or metastasis (Fig. 1A). Mice were treated with 8 Gy × 3 to the primary tumor with or without ENI and anti-CD25, which we have previously demonstrated to induce tumor eradication by depletion of Tregs when combined with SBRT[15]. We confirm in this study that anti-CD25 does deplete Tregs and that Tregs are elevated to similar levels after 8 Gy × 3

as we have seen in our past studies[15] with 10 Gy x1 (Supplementary Fig. 2B). Both tumor only and ENI irradiation resulted in similar local tumor responses, but tumor only irradiation eradicated 71% (5/7) of distant tumors compared to only 43% (3/7) of the mice treated with ENI (Fig. 1B). Compared to mice treated with SBRT or anti-CD25 alone, only mice with tumor only irradiation had a significant decrease in the amount of flank tumors that grew (Fig. 1B). To further validate the impact of ENI on distant metastasis, we used the P029 cell line, a recently created metastatic HNSCC cell line with primary pattern of spread to the lungs. Mice were similarly treated with 8 Gy × 3 to the primary tumor with or without ENI and anti-CD25. The lungs were monitored using microCT imaging for metastatic spread. Mice treated with ENI had increased primary tumor growth (Fig. 1C and Supplementary Fig. 1C). Using microCT imaging at days 41 and 47 post-tumor implantation we observed that ENI treated mice had metastatic spread to the lungs while those treated with SBRT to the tumor only had no radiographic evidence of lung metastasis (Fig. 1D). Representative microCT images of metastatic spread to the lung are shown in Fig. 1E and Supplementary Fig. 1D. This observed increase in lung metastasis of mice treated with ENI suggests that ENI accelerates metastatic growth.

As ENI is not the only means of targeting the DLNs to decrease regional and metastatic spread, we also performed surgical neck dissection in our HNSCC model. Bilateral superficial cervical lymphadenectomy was done five days post-tumor implantation on mice implanted with LY2 cells orthotopically in the buccal. At day fifteen post-tumor implantation the mice were treated with anti-CD25 and were irradiated with 10 Gy to the tumor only and were subsequently monitored for local tumor growth. Mice that received a neck dissection had an increase in local tumor growth (Supplementary Fig. 1E). These data suggest that upfront surgery followed by radiation and immunotherapy blunts the immune response and tumor control mediated by combination SBRT-immunotherapy.

To evaluate the effect of ENI on immune cell types mediating distant tumor growth, we performed flow cytometry on the blood of mice treated with or without ENI to evaluate the effects of ENI on circulating lymphocytes. While we did not find differences in the percentage of circulating CD45 cells, CD8 T cells, CD4 T cells, or NK cells (Supplementary Fig. 1F, G), we did find differences in activation of these cell types (Fig. 1F and Supplementary Fig. 1G). Most noteworthy, we found that mice implanted with the LY2 HNSCC cell line and treated with ENI had a reduction in CD8 T cells expressing CD69, an early activation marker and IL-2, a survival cytokine (Fig. 1F). Additionally, we observed differences in CD4 T cells. CD4 T cells in the mice treated with ENI had a reduction in Tbet, a transcription factor associated with a Th1 response, and CCR7, a marker associated with homing to lymph nodes and circulating memory CD4 T cells in the blood[16] (Fig. 1F). Interestingly, both mice treated with either ENI or tumor only SBRT had an increase in CD4 T cells expressing IFNg compared to anti-CD25 only treated mice (Supplementary Fig. 1G). This corroborates our findings that ENI treatment is still superior to no SBRT, or no immunotherapy shown in Fig. 1B. This, along with our data showing a reduction in distant tumor growth, led us to hypothesize that ENI is reducing a systemic immune response to therapy.

To appreciate the generalizability of these data, we tested if ENI mitigated tumor control both locally and distantly in two other tumor models: a melanoma cell line (B16-OVA) and a metastatic breast cancer cell line (4T1). Both the B16-OVA and 4T1 tumor models are known to be radioresistant. We hypothesized that these tumors may be more radiosensitive if it was ensured that no lymph nodes were included in the radiation field. We implanted these cell lines orthotopically in either the cheek skin (B16-OVA) or the mammary fat pad (4T1) to represent primary tumors. To measure distant tumor control we implanted the cell lines into the flank as well and used histology to quantitate lung metastases in the 4T1 model (Supplementary Fig. 2A).

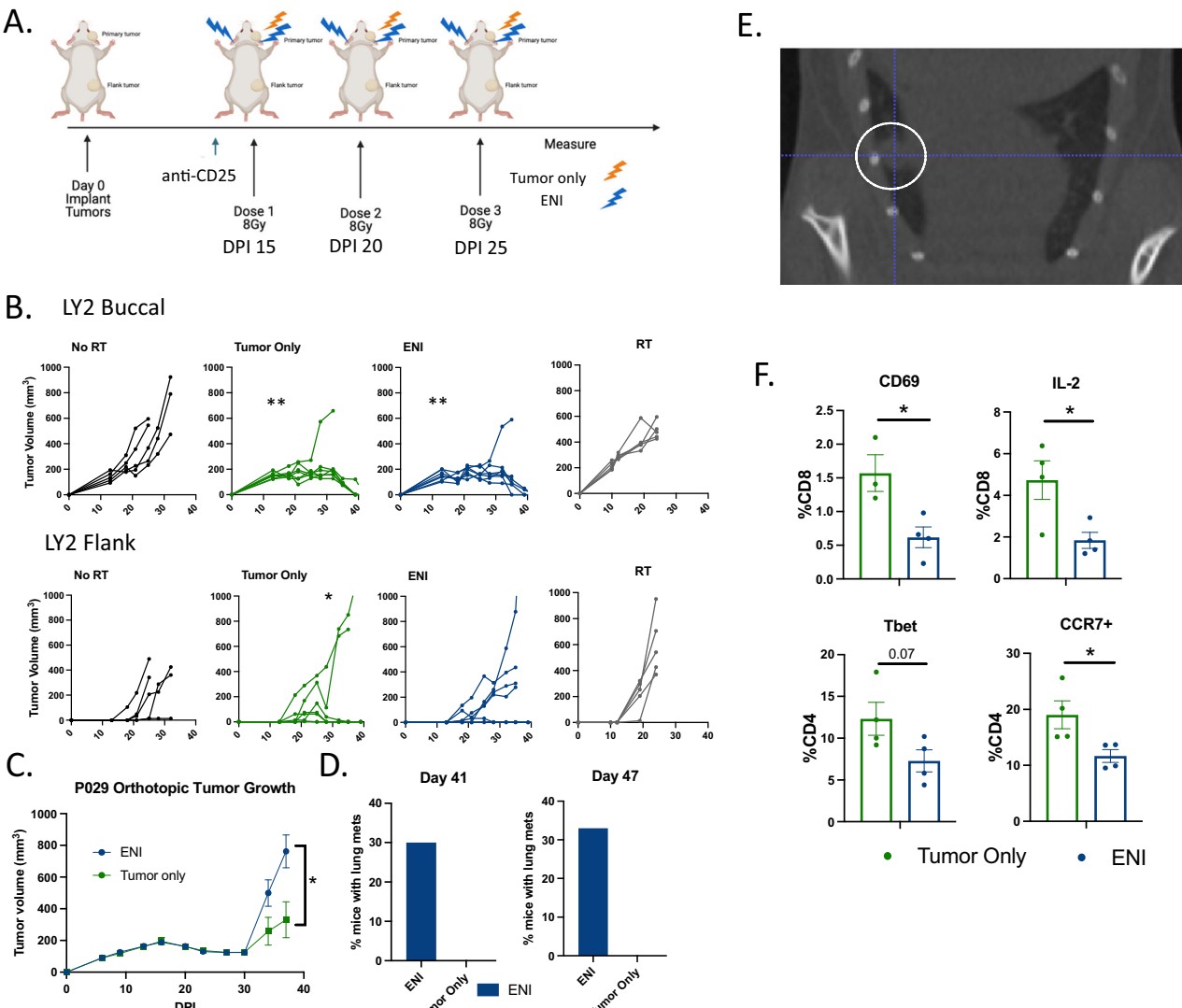

**Fig. 1 | ENI ablates the immune response to combined radiation and immunotherapy. A** Schematic of the experimental design for gross tumor irradiation with or without elective nodal irradiation (ENI). Mice were implanted both in the buccal and in the flank on day 0 post-implantation (DPI). Stereotactic body irradiation (SBRT) was given when tumors reached ~150mm$^3$ and anti-CD25 was given a day before SBRT. Created with BioRender.com. **B** Tumor growth curves, from the experiment depicted in (**A**), Buccal tumor (top) and flank tumor growth curves (bottom) for mice treated with anti-CD25 (*n* = 5), anti-CD25 and tumor only SBRT (*n* = 7), anti-CD25 and ENI (*n* = 7), and tumor only SBRT alone (*n* = 5). **C** Buccal tumor growth curves for mice implanted with the P029 cell line (*n* = 10 per group). Mice were implanted in the buccal on day 0 post-implantation (DPI). SBRT was given when tumors reached ~150mm$^3$ and anti-CD25 was given a day before SBRT and once a week thereafter. The doses of SBRT were spaced by 4–5 days. **D** Quantification of the percentage of mice with P029 tumors that had radiographically detectable lung metastases at days 41 (ENI, *n* = 10; tumor only, *n* = 7) and 47 (ENI, *n* = 9; tumor only, *n* = 7) post-tumor cell implantation. Lung metastases were evaluated by microCT images. **E** A representative microCT image of a lung metastasis identified in a mouse treated with ENI in the P029 model. A metastasis is highlighted with a white circle. **F** Flow cytometry analysis of blood taken from mice at day 24 DPI in the experiment depicted in (**A**) (ENI, *n* = 4; tumor only, *n* = 4). CD8 T cells were defined as CD45+CD3+CD8+ and CD4 T cells were defined as CD45+CD3+CD4+. For tumor growth at different time points, 3 or more groups differences were determined by a One-Way ANOVA test with Tukey's post hoc comparisons, with only 2 groups a Two-Way ANOVA was used. To test if there is a difference between tumor only SBRT and ENI treatment groups in reducing the number of mice that grew flank tumors, we used a Fischer's Exact test. For the flow cytometry analysis, a two-tailed student's t-test was used. Significance was determined if the p-value was <0.05* and <0.01**. The error bars represent the standard error of the mean (± SEM). Source data are provided as a Source Data file. *p*-values are indicated for figures **B**. buccal tumor only **0.0035, buccal ENI **0.0032, *0.0278, (**C**) *0.0371, and (**F**) CD69 *0.0222, IL-2 *0.0284, CCR7 *0.0366.

Mice implanted with B16-OVA melanoma cells were treated with ENI, bilateral neck irradiation, had an increase in primary tumor and distant tumor growth (Supplementary Fig. 2B). Similarly, mice treated with ENI, bilateral axillary and inguinal node irradiation, implanted with the 4T1 cell line had a significant increase in distant flank tumor growth (Supplementary Fig. 2C) and increased lung metastasis. Metastasis in the lungs was visualized with microCT and quantified with H&E staining (Supplementary Fig. 2D, E). We also repeated these experiments using anti-PD-1 as an immunotherapy in combination with radiation in

the 4T1 model. We observed similar trends in the 4T1 model where the flank tumor growth was decreased the most in the mice treated with tumor only radiation compared to ENI (Supplementary Fig. 2F).

**ENI decreases antigen-experienced T cell expansion in the DLNs and infiltration into the TME**
To determine if the effectiveness of tumor only irradiation combined with anti-CD25 was dependent on CD4 and/or CD8 T cells, we pharmacologically depleted either CD4 T cells, CD8 T cells, or both cell

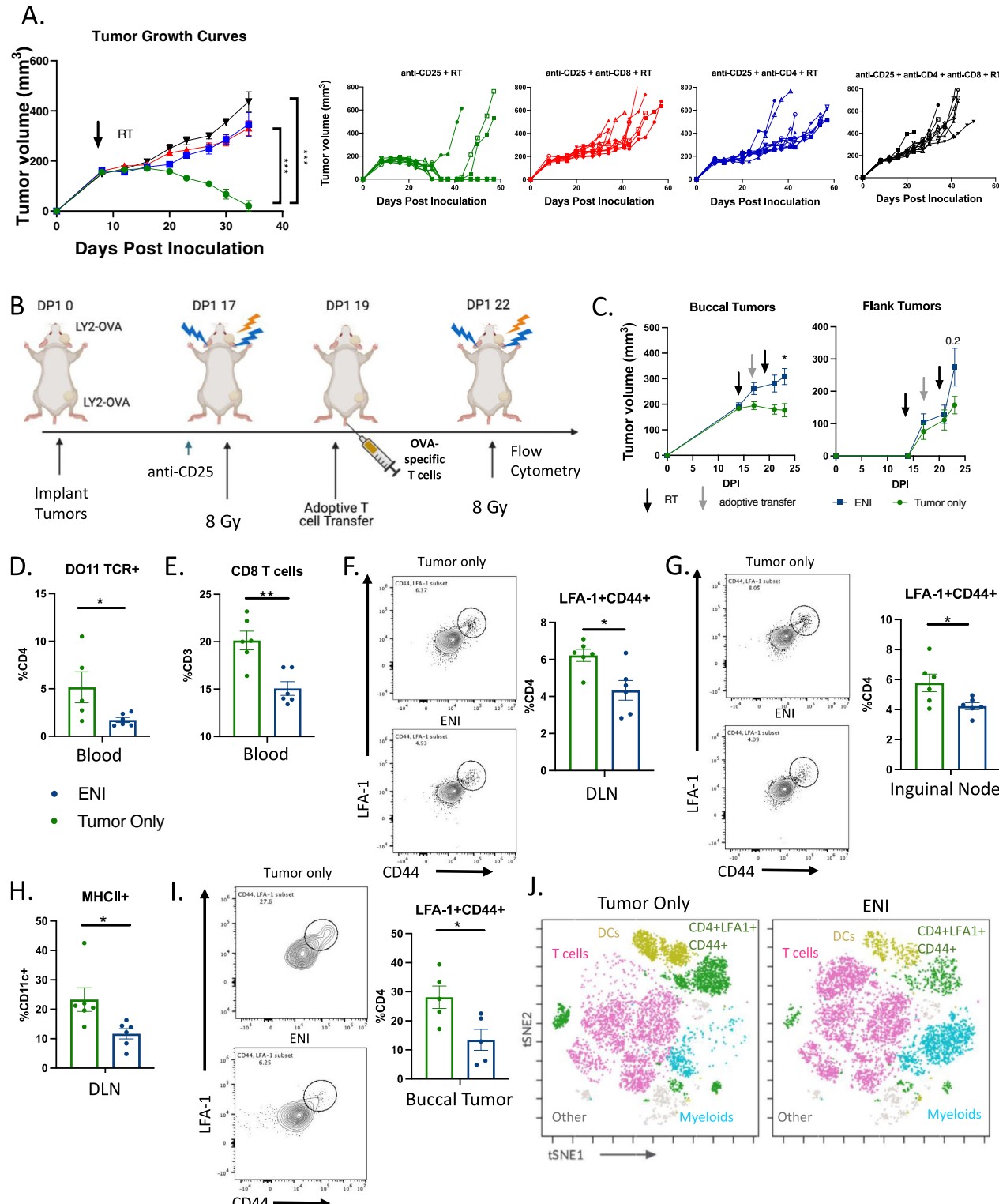

types. We observed that if either or both cell types are depleted, the mice were unable to eradicate their tumors (Fig. 2A) suggesting that combination SBRT with immunotherapy is dependent on both CD4 and CD8 T cell activation. Since priming of T cells occurs in the DLNs, where most of the tumor-antigen specific T cells reside[17], we hypothesized that ENI was reducing tumor-antigen specific T cell priming and consequently reducing circulating antigen-specific T cells.

To examine if ENI reduces antigen-specific T cell expansion and circulation, we generated an ovalbumin expressing cancer cell line (LY2-OVA) and implanted these cells into both the buccal and the flank.

After the first dose of 8 Gy, we adoptively transferred OVA-specific T cells, isolated from DO11 mice, into the mice with LY2-OVA implanted tumors (Fig. 2B). The mice received radiation with or without ENI in two fractions of 8 Gy before flow cytometry was conducted. Similar to our previous experiment using the non-antigen specific LY2 cell line, we observed a decrease in distant tumor growth in the LY2-OVA cell line. However, using the LY2-OVA cell line we now also observed a decrease in the primary tumor growth in tumor only treated mice compared to mice treated with ENI concordant with our findings in the B16-OVA model and the PO29 model (Fig. 2C). This may suggest that

**Fig. 2 | ENI decreases antigen-experienced T cell expansion in the DLNs and infiltration into the TME. A** Tumor growth curves of mice depleted of CD4 T cells, CD8 T cells, both CD4 and CD8 T cells, or neither before tumor implantation ($n = 10$ for all groups). Mice with LY2 tumors were treated with 10 Gy × 1 SBRT to the tumor only when the tumors reached ~200mm³ and treated with anti-CD25. **B** Schematic of experimental design. OVA-specific T cells ($1 × 10^5$ cells) were adoptively transferred into the mice via tail vein injection after the first dose of SBRT. At day 23 post-tumor cell implantation tumors, DLNs, and blood were harvested for flow cytometry. Created with BioRender.com. **C** Buccal and flank tumor growth curves for the experiment described in (**B**) (tumor only, $n = 7$; ENI $n = 10$). **D** Quantification of the percentage of CD4 T cells (CD45+CD3+CD4+) in the blood that were also DO11 TCR+ (tumor only, $n = 5$; ENI $n = 6$). **E** Quantification of CD8 T cells (CD45+CD3+CD8+) in the blood (tumor only, $n = 6$; ENI $n = 6$). **F** Flow plots and quantification of CD4+LFA−1+CD44+ T cells in the primary tumor DLN (tumor only, $n = 6$; ENI $n = 6$). **G** Flow plots and quantification of LFA1+CD44+CD4 T cells in the inguinal node (tumor only, $n = 6$; ENI $n = 6$).

**H** Quantification of DCs (CD45+CD3−CD11c+MHCII+) in the primary tumor DLN (tumor only, $n = 6$; ENI $n = 6$). **I** Flow plots and quantification of LFA-1+CD44+ T cells in the buccal tumor (tumor only, $n = 5$; ENI $n = 5$). **J** t-SNE with FlowSOM population overlay of multi-spectral flow cytometry data (tumor only, $n = 6$; ENI $n = 6$). T cells (pink) were defined by expression of CD3 and CD4 or CD8, dendritic cells (DCs) (gold) were defined by CD3−CD11c+MHCII+ cells, CD4+LFA−1+CD44+ T cells (green) were defined by expression of CD4, LFA−1 and CD44, and myeloid cells (light blue) was defined by CD3−CD11c− cells. For tumor growth at selected time points, 3 or more groups differences were determined by a One-Way ANOVA test with Tukey's post hoc comparisons, with only 2 groups a mixed-effects model was used. For the flow cytometry analysis, a two-tailed student's *t* test was used. Significance was determined if the *p*-value was <0.05*, <0.01**, and <0.001***. The error bars represent the standard error of the mean (±SEM). Source data are provided as a Source Data file. *p*-values are indicated for figures **A.**\*\*0.0014, CD8 depletion \*\*\*0.0002, double depletion \*\*\*0.0001 **C**. *0.0183, **D**. *0.0478, **E**. \*\*0.002, **F**. *0.0125, **G**. *0.0366, **H**. *0.024, and **I**. *0.252.

the degree of antigenicity of the tumor type impacts the degree to which ENI affects tumor growth, affecting both local and distant tumor growth. However, regardless of antigenicity, and across different tumor models, ENI appears to be uniformly deleterious.

To determine if ENI decreases antigen-specific T cells systemically, we looked for OVA-specific CD4 T cells in the blood. We found that ENI decreased the amount of OVA-specific CD4 T cells in the blood (Fig. 2D). Representative gating for OVA-specific CD4 T cells is shown in Supplementary Fig. 3A. We also observed a decrease in circulating CD8 T cells in mice treated with ENI (Fig. 2E). Given that the DLNs act as the site of T cell priming[18,19], we next examined the effects of ENI on T cell priming within the DLN compartment using flow cytometry. We observed a decrease in LFA-1+CD44+CD4 T cells in the DLN in mice treated with ENI, both markers that represent early activation/priming markers on antigen-experienced CD4 T cells[20] (Fig. 2F and Supplementary Fig. 3B). Even distantly in the inguinal lymph node, the DLN of the flank tumor, we observed a decrease in T cell priming via LFA-1+CD44+CD4 T cells in mice treated with ENI (Fig. 2G). Along with a decrease in T cell priming in both primary and distant lymph nodes of ENI mice, we observed that mice treated with ENI had a decrease in dendritic cells (DCs) (CD11c+MHCII+ cells) in the primary tumor DLNs (Fig. 2H). This decrease in T cell priming and abundance of DCs did not translate to a decrease in cellularity in the DLNs or the size of the DLNs (Supplementary Fig. 3C). Next, we aimed to determine the effects of ENI on the primary (local or buccal) TME. Treatment with ENI decreased LFA-1+CD44+CD4 T cells in the TME of the buccal tumor, indicating a decrease in the presence of antigen-experienced T cells locally (Fig. 2I, J). This was also supported by an overall decrease in T cells in the buccal tumor in the ENI treated mice (Supplementary Fig. 3D). Altogether these data suggest that ENI dampens the immune response generated by SBRT and immunotherapy by reducing antigen-experienced T cells systemically.

## Systemic, long-term, DLN-independent memory is formed with tumor only SBRT

We have previously demonstrated that SBRT to gross tumor only combined with anti-CD25 does confer antigen specific memory[15]. However, whether this treatment triggers local memory or systemic memory remains to be validated. To test this, we rechallenged cured mice, ones originally implanted with only a buccal tumor and treated with tumor only SBRT, by re-implanting the same cell line (LY2) in both the buccal and in the flank. Compared to naïve mice, we observed no tumor growth in either location (Fig. 3A), suggesting that cured mice have systemic immune memory. Next, to examine which immune cell types are driving the lack of tumor cell growth upon rechallenge, we rechallenged mice again but instead harvested blood and DLNs 4 days after buccal implantation rechallenge to capture the developing adaptive memory immune response (Supplementary Fig. 4A). We

observed major shifts in immune cell populations between the cured mice and the naïve mice in the DLNs and the blood (Fig. 3B), suggesting large differences in systemic immune cell activation in all observed immune cell populations (CD8 T cells, CD4 T cells, NK cells, and MHCII + cells).

Next, we examined how the changes in the blood compartment compared to those in the DLNs. Concordant with our previous findings (Fig. 1), Th1 CD4 T cells and CD8 T cells were markedly activated in the blood and in the lymph nodes (Fig. 3C–E). Confirming a strong CD4 T cell activated phenotype within the DLNs was the significant increase in CD44+, IFNg+, and IL2+ cells (Fig. 3E and Supplementary Fig. 5A–C) as well as an increase in a CD4 Th1 signature population (Fig. 3F). When in circulation, this CD4 Th1 signature population has been shown to correlate with response to immunotherapy[21]. Unlike the blood compartment, there was an increase in T cell replication in the DLNs (Fig. 3G and Supplementary Fig. 5D). Correlating with the increase in T cell activation and replication in the DLNs was an observed increase in activated cDC1 DCs that homed to the DLNs (Fig. 3H). This was manifested by the increased expression in CD103, CD80, Ki-67, and CCR7 on DCs (Fig. 3H). These findings are suggestive of increased DC-mediated priming and T cell expansion within the DLN resulting in an increase in circulating effector T cells, derived from memory cells, in cured mice upon re-challenge.

To further understand what cell types are circulating and responding to re-challenge in cured mice vs. naïve mice, we compared changes in circulating immune populations between naïve and cured mice at two time points, before (baseline) and after re-challenge (Day 4). We observed an increase in circulating activated and replicating CD4 T cells, defined as IL-2 producing and Ki-67 expressing cells compared to baseline in cured mice (Fig. 3I, J). CD8 T cell activation increased between baseline and post-rechallenge in both naïve mice and cured mice (Fig. 3K). No significant differences in NK cell activation in the blood was observed between cured mice and naive mice (Supplementary Fig. 4B). However, a significant increase in circulating MHC II expressing CD11c⁺ cells, suggesting that there is an increase in DCs in the blood of cured mice (Fig. 3L). We confirmed that these findings were also true in older mice (>1.5 years old) (Supplementary Fig. 4C) suggesting that memory is maintained long term and even in advanced age. To confirm that the memory response was dependent on CD4 T cells and CD8 T cells, we depleted these cell types in two groups of cured BALB/c mice and once again rechallenged them (Supplementary Fig. 4D). We observed tumor growth in both groups suggesting that both CD4 and CD8 T cells are responsible for the adaptive memory immune response observed (Fig. 3M).

With the understanding that mice treated with SBRT and anti-CD25 develop systemic immunity, we aimed to determine if systemic memory was maintained in the DLNs or if there was a circulating/ systemic population of memory cells. To this end, we completed a

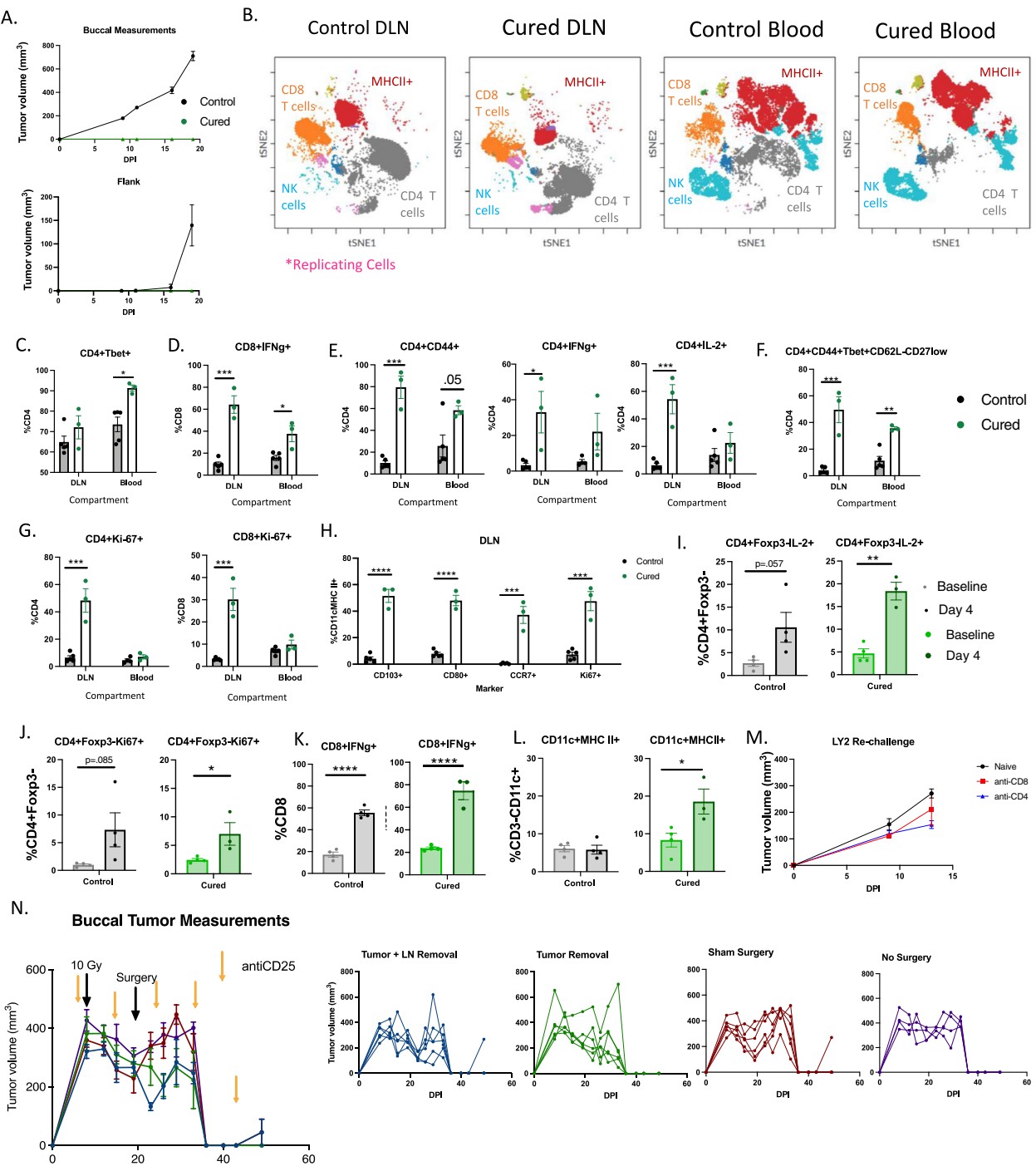

**Fig. 3 | Systemic, long-term, DLN-independent memory is formed with tumor only SBRT. A** Tumor growth curves of mice rechallenged with LY2 tumor cells in both the buccal and in the flank (Cured, *n* = 3; Naïve, *n* = 8). **B** t-SNE clustering and FlowSOM clusters superimposed of CD45+ immune cells in the DLN and Blood. CD8 T cells (CD3+CD8+), CD4 T cells (CD3+CD4+), NK cells (CD3-NKp46+), replicating cells (Ki-67+), and MHCII+ cells are highlighted. **C**–**H** Quantification of activation markers expressed by T cells in the DLN and in the blood of naïve mice or mice rechallenged with LY2 cells in the buccal (<6 months after initial eradication) (Cured, *n* = 3; Naïve, *n* = 5). **I**–**L** Quantification of immune cells in the blood of mice before and after rechallenging with LY2 cells in the buccal or implanting LY2 cells into naïve mice (<6 months after initial eradication) (Cured, *n* = 4; Naïve, *n* = 4). **M** Tumor growth curves of naïve mice or cured mice treated with either a CD8 or CD4 T cell depleting antibody (anti-CD4, *n* = 5; anti-CD8, *n* = 3; naïve, *n* = 5).

**N** Tumor growth curves of mice treated with neoadjuvant SBRT and anti-CD25 with a neck dissection occurring on day 19 post-tumor implantation. Individual tumor growth curves are provided to the right (Tumor and LN removal, *n* = 6; Tumor removal, *n* = 6; sham surgery, *n* = 6; no surgery, *n* = 4). A two-tailed student's *t* test was used to examine group differences. Significance was determined if the *p*-value was <0.05*, <0.01**, <0.001***, and <0.0001****. The error bars represent the standard error of the mean (±SEM). Source data are provided as a Source Data file. *p*-values are indicated for figures **C.** *0.011, **D.** ***0.0002, *0.0114, **E.** CD4+CD44****0.0001, *0.0137, CD4+IL-2+ ***0.0009, **F.** ***0.0008, **0.0014, **G.** CD4**0.0007, CD8**0.0003, **H.** CD103**** < 0.0001, CD80**** < 0.0001, CCR7***0.0006, Ki-67***0.0004, **I.** **0.0011, **J.** *0.0423, **K.** control ****<0.0001, cured ****<0.0001, and **L.** *0.0341.

neck dissection approximately 2 weeks after SBRT and anti-CD25 treatment, a timepoint after initial response and after memory formation, and implanted flank tumors in mice post-operatively to examine if this would influence distant tumor growth (Supplementary Fig. 4E). As expected, there was no difference in local tumor growth as the initial immune response had already primed (Fig. 3N). There was also no difference in distant tumor growth between the two groups suggesting that tumor-specific T cells maintain a circulating memory population capable of preventing distant tumor growth weeks after treatment (Supplementary Fig. 4F). We repeated this experiment and preformed flow cytometry on day 30 post-tumor implantation on both the blood to better understand if there were any circulating immune cell differences in the immune system resulting from a neck dissection. Again, we did not see any differences in primary tumor growth (Supplementary Fig. 4G). We also observed no meaningful immune cell differences in the blood (Supplementary Fig. 4H) as expected based on our tumor growth findings, suggesting that after initial priming, systemic memory is formed and not reliant on the DLNs. Our findings that the effectiveness of SBRT and immunotherapy is decreased by ENI and upfront surgery of DLNs, but unaltered by late neck dissection, suggest the DLNs are responsible for initial priming and expansion. Specifically, the lack of a detrimental effect of late neck dissection on either local or distant tumor growth suggests that after initial priming, a circulating anti-tumor memory T cell population is generated and is sufficient to eradicate distant tumor growth. These data collectively imply that when surgery is delayed and sequenced significantly after radiation and immunotherapy, after a sufficient systemic immune response has been generated, it carries no negative effects on local or distant metastasis.

## ENI reduces epitope spreading and T cell activation in the distant DLN and tumor

Metastatic tumors are often not a homogeneous population and can differ significantly from the original primary tumor. This is thought to be due to a small population of tumor cells breaking off from the primary tumor and repopulating a new TME niche resulting in subclonal expansion[22]. Given the observed reduction in distant tumor growth in our metastatic models of HNSCC and breast cancer, we hypothesized that ENI combined with immunotherapy may reduce epitope spreading. Epitope spreading is a described phenomenon when an immune response generated against a certain antigen or antigens is able to promote antigen-presenting cell (APC) presentation of a different antigen or antigens to T cells, stimulating expansion of new antigen specific T cells[23]. To model a different antigen repertoire between the primary tumor and the flank tumor, we implanted LY2 tumors in the buccal and LY2-OVA tumor cells in the flank (Fig. 4A). Once the tumors approached 200mm³, the mice were started on immunotherapy and the following day administered a dose of 8 Gy SBRT as done previously. The mice received antigen-specific T cells two days later and a final dose of 8 Gy SBRT three days after the adoptive transfer of T cells. Consistent with our previous findings (Fig. 1B), we observed no difference in the primary tumor growth, but a significant difference was observed in distant tumor growth (Fig. 4B).

Since we only implanted OVA-LY2 cells in the flank tumor, if treating mice with ENI was reducing epitope spreading, we would expect a reduction in OVA-specific CD4 T cells circulating in the blood. We found that there was indeed a reduction in circulating OVA-specific CD4 T cells in mice treated with ENI (Fig. 4C and Supplementary Fig. 6A). As an increase in OVA-specific T cells in the blood should be a result of T cell priming in the DLN of the flank tumor, we investigated if there was an increase in T cell activation in the inguinal lymph node, the DLN of the flank tumor. We found that mice treated with ENI had a reduction in Th1 CD4 T cells (Tbet+) that were OVA-specific in the inguinal lymph node (Fig. 4D). Additionally, we found overall fewer Th1 CD4 T cells in the inguinal lymph node of mice treated with ENI,

defined as IFNg and IL-2 producing cells, which represented the majority of Tbet+ T cells (Fig. 4E). We also observed a decrease in Granzyme B production in CD8 T cells in the inguinal lymph node of mice treated with ENI (Fig. 4F). To determine if ENI was reducing the ability of T cells to migrate from the DLNs to the TME, we examined the expression of migration markers on T cells in the primary DLNs. We found that ENI treated mice had less expression of CCR7 on CD4 T cells. This suggests that either fewer CD4 T cells are being recruited to the DLNs or that CD4 T follicular helper cells are not being maintained in the DLNs to activate B cells[24] (Supplementary Fig. 6B). We also observed that CD8 T cells in the DLN of mice treated with ENI have decreased CXCR3 expression, suggesting that they are less likely to migrate from the DLN to the TME[24] (Supplementary Fig. 6C). We hypothesized that treatment with ENI would decrease T cell priming through antigen specific T cell and/or DC cell death due to RT. ENI treated mice did indeed have a greater percentage of OVA-specific CD4 T cells and DCs undergoing apoptosis (determined by expression of cleaved caspase 3) in the DLNs (Fig. 4G). An increase in apoptosis was not seen in the inguinal nodes between the two groups and overall and there was little to no apoptosis present in the inguinal nodes (Supplementary Fig. 6D). Finally, we examined T cell infiltration into the TME of the flank tumor. Mice treated with ENI had fewer T cells per mg of tumor compared to mice with tumor only treatment (Fig. 4H). Of the CD4 T cells that managed to infiltrate the flank tumor in the ENI mice, fewer were OVA-specific compared to mice treated with tumor only SBRT (Fig. 4I). Similarly, examining all T cells, tumor only treated mice had an increase in activated CD8 T cells and CD4 T cells in the flank tumor TME (Fig. 4J). These data collectively suggest that even if the distant metastasis expresses different tumor antigens from the primary tumor, if the primary and distant tumor share some antigens, epitope spreading can occur resulting in an increase in T cells in the distant tumor and additional RT to the distant tumor is unnecessary. It also suggests that ENI reduces epitope spreading and subsequent priming at distant tumor sites.

## Sentinel node resection, or irradiation, reduces regional recurrence

Although tumor only SBRT combined with immunotherapy produces a robust local, circulating, and distant anti-tumor response, we still witnessed regional recurrence in these mice. In the experiment depicted in Fig. 1A, three out of the four mice not treated with ENI, which cleared both local and distant tumors, developed regional metastases while none of the mice treated with ENI developed regional metastases. We also observed an increase in regional metastasis in the mice that did not receive a neck dissection from the experiment depicted in Fig. 3N. Mice that underwent a neck dissection had similar local recurrence free survival as those that did not, but the mice that did not have a neck dissection had increased regional recurrence that decreased their overall survival rates (Fig. 5A).

To examine if distant tumor implantation was influencing the rate of regional recurrence, we implanted mice with a buccal tumor or with both a buccal and a flank tumor and treated them with one dose of 10 Gy SBRT and anti-CD25 (Fig. 5B). We did not observe any differences in either local tumor growth or rate of regional recurrence between animals implanted with a buccal tumor and those implanted with both a buccal and a flank tumor (Supplementary Fig. 7A, B). Despite being able to clear either a local buccal tumor or both a local and distant flank tumor, mice treated with tumor only irradiation still developed regional metastases (Supplementary Fig. 7C). At 150 days follow up, we observed that regional metastasis was the dominant pattern of failure in these mice (Fig. 5C). We confirmed through histology that these regional metastases were indeed derived from the primary HNSCC tumor cell line and not a leukemia or lymphoma (Supplementary Fig. 7D). We also confirmed that tumor only treated mice had evidence of cancer cells within the sentinel lymph node after treatment, while

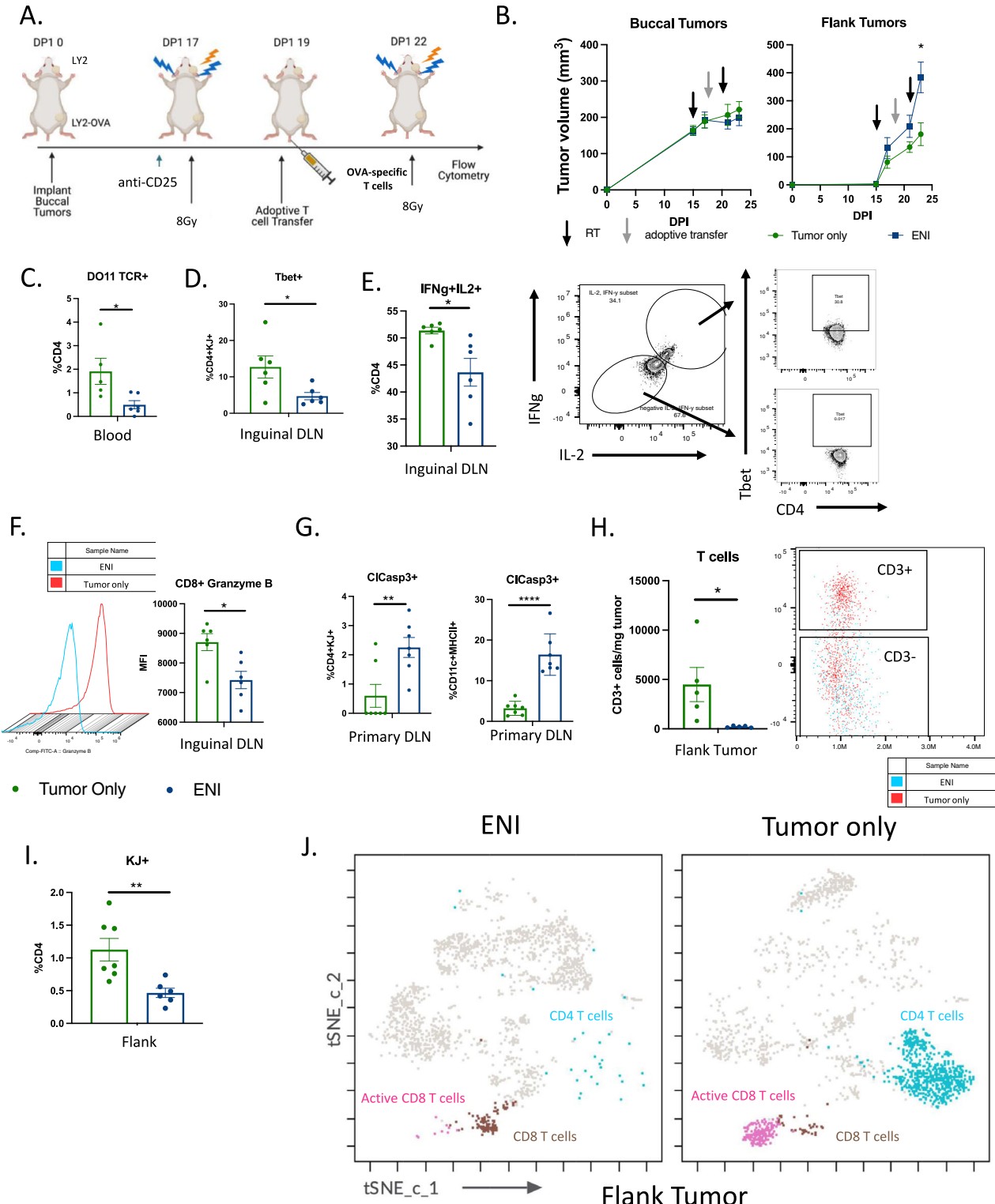

mice treated ENI did not (Supplementary Fig. 7E). Based on these findings, we repeated our initial experiment depicted in Fig. 1A, but without a flank tumor so that a greater percentage of ENI mice would survive to observe regional recurrence. We found that no mice treated with ENI that eradicated the primary tumor had regional recurrence (0/9), while 4/8 mice that eradicated the primary tumor that were treated with tumor only SBRT had regional recurrence (Fig. 5D). This further suggested that ENI reduces regional recurrence, as is has been known to do in the clinical setting.

The regional recurrence following tumor only SBRT and immunotherapy combination treatment was an unexpected observation especially since this occurred in the context of both local and distant tumor eradication and despite evidence showing systemic immune memory formation (Fig. 3). Based on our results that upfront neck dissection is detrimental for a local tumor immune response (Supplementary Fig. 1E), but that removal of the nodes after SBRT and anti-CD25 did not dampen the immune response (Fig. 3N), we hypothesized that removal of the sentinel lymph nodes after SBRT and anti-CD25

**Fig. 4 | ENI reduces epitope spreading and T cell activation in the distant DLN and tumor. A** Schematic of experimental design. An adoptive transfer of DO11 CD4 T cells ($1 \times 10^5$ cells) was given to the mice two days after the first dose of SBRT. Tissue was harvested three days after the last dose of SBRT. Created with BioRender.com. **B** Tumor growth curves of the mice depicted in (**A**) (tumor only, $n = 9$; ENI, $n = 10$). **C** Quantitation of the percentage of CD4+ T cells that were positive for the DO11 TCR (KJ+) in the blood (ENI, $n = 6$; tumor only, $n = 5$). **D** Quantification of Tbet+ antigen-specific CD4 T cells (DO11 TCR+ or KJ+) in the inguinal DLN (ENI, $n = 6$; tumor only, $n = 6$). **E** Quantitation of IFNg+IL-2+CD4 T cells in the inguinal lymph node (ENI, $n = 6$; tumor only, $n = 6$). Gating strategy for IFNg+IL-2+ cells are provided on the right. Representative flow plots of Tbet expression in IFNg+IL-2+ and IFNg-IL-2- cells are also shown. **F** Quantitation of the mean florescent intensity (MFI) of Granzyme B in CD8 T cells in the inguinal lymph node. Histogram of MFI is provided to the left (ENI, $n = 6$; tumor only, $n = 6$). **G** Quantification of cleaved caspase 3 expression on KJ+CD4 T cells and on DCs (CD11c+MHCII+) in the DLNs (ENI, $n = 7$; tumor only, $n = 7$). **H** Quantification of the number of CD3+ T cells in the flank tumor. Flow plot of CD3+ cells are provided to the right (ENI, $n = 5$; tumor only, $n = 5$). **I** Quantification of KJ+CD4 T cells in the flank tumor (ENI, $n = 6$; tumor only, $n = 6$). **J** t-SNE with FlowSOM population overlay of CD45+ cells (ENI, $n = 6$; tumor only, $n = 6$). CD8 T cells (brown) were defined as having CD3 and CD8 expression. Activation CD8 T cells (pink) were defined as having expression of CD3, CD8, IFNg and Granzyme B expression. CD4 T cells (blue) were defined as having CD3 and CD4 expression. For tumor growth at selected time points, treatment difference was determined by a Two-Way ANOVA test. For the flow cytometry analysis, a two-tailed student's $t$ test was used. Significance was determined if the p-value was <0.05\*, <0.01\*\*, and <0.001\*\*\*. The error bars represent the standard error of the mean (± SEM). Source data are provided as a Source Data file. p-values are indicated for figures **B.** \*0.015, **C.** \*0.0260, **D.** \*0.0313, **E.** \*0.015, **F.** \*0.011, **G.** \*\*0.0077, \*\*\*\*<0.0001, **H.** \*0.0391, and **I.** \*\*0.0068.

treatment would preserve the local immune response while reducing regional recurrence rate. Sentinel lymph nodes were determined by the presence of an enlarged anterior cervical lymph node or nodes on the ipsilateral side at time of surgery which have been the location of regional metastasis observed in prior experiments. To test this hypothesis, mice were treated with tumor only radiation and then three days later a sentinel lymph node resection was performed (Supplementary Fig. 7F). While the rate of local recurrence was similar between the groups (1/10 vs. 2/10) (Fig. 5E), significantly different rates of regional recurrence was observed. No regional metastases were observed in the sentinel lymph node resection group by day 150 compared to 8/15 in mice that did not receiving sentinel lymph node resection (Fig. 5F). As we also observed that ENI can reduce regional recurrence, we asked if sentinel lymph node irradiation is sufficient to eliminate regional recurrence. A schematic of how sentinel lymph node radiation was conducted is provided in Supplementary Fig. 7G. We found that mice treated with sentinel lymph node irradiation, that eradicated the primary tumor, had no regional recurrence (0/9) (Fig. 5D). Altogether these data support the notion that the timing and extent of nodal resection and/or irradiation play a role in local and regional recurrence. These findings also support the hypothesis that although a systemic immune response was generated, one sufficient to eradicate local and distant tumors, tumor-specific immune cells are either unable to enter regional metastases or are inactivated upon entry.

### Tumor only SBRT increases immune responses in canines and humans with HNSCC

To determine if tumor only irradiation would produce durable anti-tumor immune responses in non-murine models, we evaluated circulating lymphocytes in mice and dogs treated with ENI versus tumor only SBRT. In our murine models, we observed that lymphocyte numbers decreased while granulocytes and monocytes increased in mice treated with ENI compared to tumor only treated mice (Fig. 6A and Supplementary Fig. 8A). ENI also reduced white blood cell and lymphocyte counts compared to sentinel lymph node irradiated mice while sentinel lymph node removal did not reduce white blood cell counts (Fig. 6B, Supplementary Fig. 8B). These data suggest that reducing the amount of lymph nodes irradiated increases circulating lymphocytes.

To validate these findings in non-murine models, we interrogated data from a Phase I clinical trial in canine patients that received tumor only irradiation versus ENI (Fig. 6C) and from a recently completed phase I/Ib human clinical trial of neoadjuvant SBRT with durvalumab in patients HPV-unrelated locally advanced oral cavity HNSCC (NCT03635164) that received tumor only irradiation. Data from the canine patients showed that compared to tumor only SBRT, ENI treatment reduced CD4 and CD8 T lymphocyte counts in nasal lavages which have been shown to be representative of the TME[25] (Fig. 6D). Temporal data from human patients further substantiated the finding that, in contrast to historical controls with prolonged conventional RT

that more often than not includes bilateral lymph node irradiation[26–29], this trial shows that tumor only SBRT does not reduce circulating lymphocytes. The trial was designed such that patients did not receive ENI and received delayed neck dissection 3–6 weeks after treatment had been concluded (Fig. 6E). In this phase I/Ib clinical trial we found that patients that responded to treatment had increased levels of T cells after SBRT (Fig. 6F). These findings corroborated results of our mouse studies suggesting that ENI dampens the ability of the immune system to respond to SBRT and immunotherapy by reducing systemic CD4 and CD8 T cells.

To appreciate the impact of avoiding irradiation of elective nodes, we examined non-irradiated nodes taken from patients in the phase I/Ib human clinical trial post-SBRT and immunotherapy (SBRT-IO) and compared them to normal non-treated control patients. By avoiding ENI, non-irradiated lymph nodes showed activated T cells, defined by IFNg expression, in the DLNs at time of surgery compared to normal nodes taken from non-treated patients that showed no activation (Fig. 6G). Even 3–6 weeks out from treatment, lymph nodes that are routinely removed at time of surgery remain active and potentially priming tumor-specific T cells. To further corroborate these data, we examined RNA sequencing of patient tumors before treatment and post-SBRT-IO. We used a machine learning algorithm to predict the amount of TCR activation in the tumor post-treatment at the time of lymph node resection. We found that post-SBRT-IO treatment there was still an increase in TCR activation in the TME (Fig. 6H). This increase in TCR activation was matched by an increase in CD8 T cells (Fig. 6I) and a decrease in cancer cells in the TME (Fig. 6J).

To examine the effect of ENI on antigen presentation in canine patients, we preformed RNA sequencing using nanostring technology to sequence LNs of dogs two weeks post-SBRT that were treated with and without ENI. We interrogated the following antigen presentation genes; MHC II (*DLA-DRA, DLA-DMA, DLA-DQB1, DLA-DMB, DLA-DQA1, DLA-DOB*), MHC I (*B2M*), and co-stimulatory molecules (*CD40* and *CD80*) (Supplementary Fig. 8C). All these genes associated with an increase in antigen presentation (MHC I and/or MHC II antigen presentation) were increased in dogs that received ENI. Activation genes associated with effector T cells (*GZMB* and *IFNG*) and *CXCL10*, a cytokine associated with increased T cell homing, were also increased in ENI treated dogs (Supplementary Fig. 8C). On the other hand, genes associated with immunosuppression (*FOXP3, IL10RA,* and *IL17RB*) were decreased in dogs treated with ENI (Supplementary Fig. 8C). However, none of the genes associated with antigen presentation, T cell activation, or immunosuppression were significantly different between dogs that received ENI and those that did not (Supplementary Fig. 8). Although we did not find any significant differences in gene expression in the LN, irradiation of the lymph nodes translated into a decrease in T cells in the TME three days post-RT (Fig. 6D). Altogether these data suggest that ENI reduces systemic T cells in both mouse models and canine HNSCC patients and that avoiding treatment with ENI in human

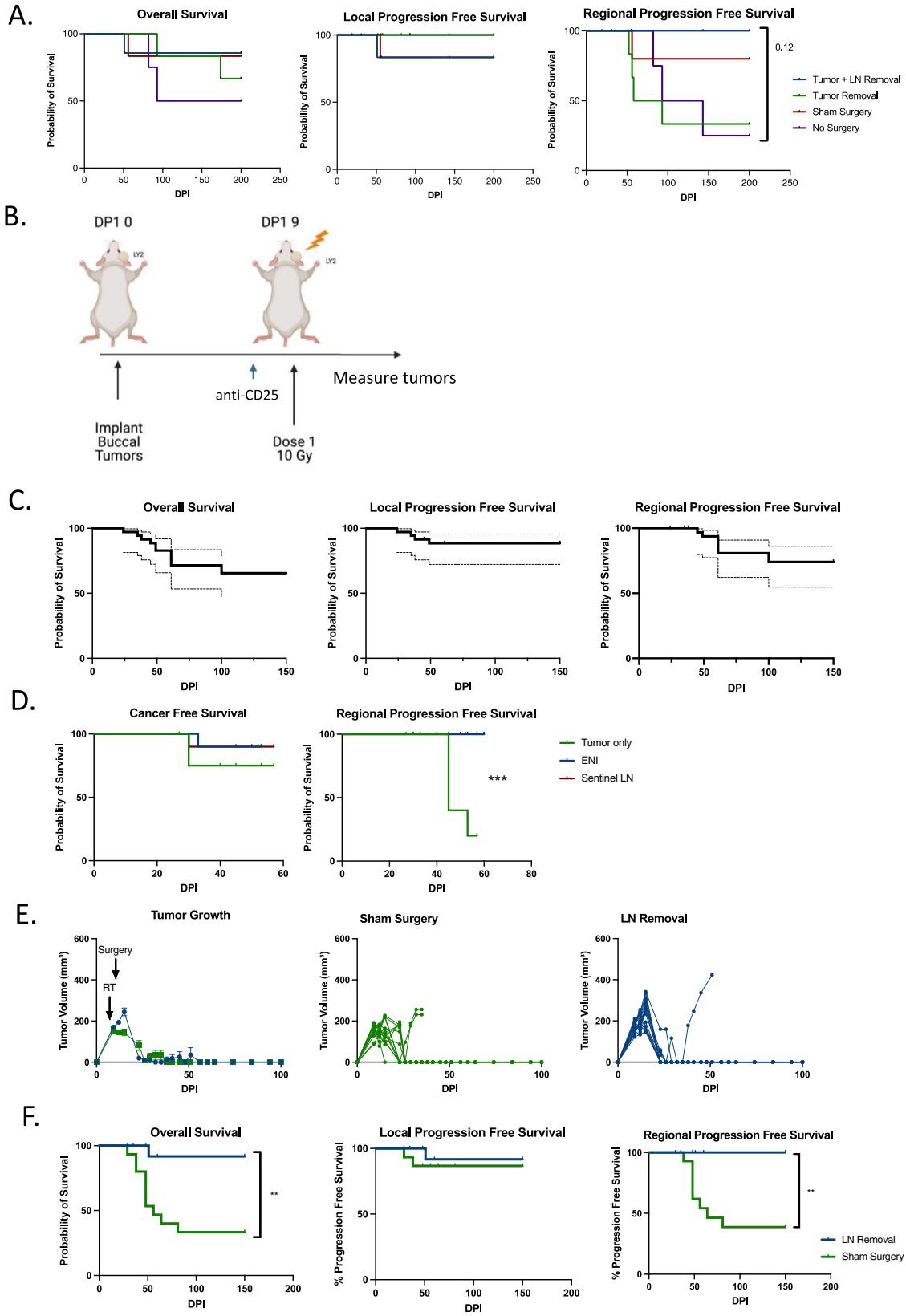

patients with HNSCC induces a robust immune response is generated similar to what we observed in mice.

## Discussion

As the importance of the immune system is realized in cancer therapy, it has become imperative to integrate new knowledge with common practices. ENI is a good example of how our understanding of cancer therapy has evolved over time. ENI is applied in standard of care therapy for HNSCC to "sterilize" microscopic disease based on pathological probability of presence of cancer cells within certain lymph node stations by histological analysis of surgical specimens. In that context, ENI was a logical approach in reducing microscopic

**Fig. 5 | Sentinel node resection, or irradiation, reduces regional recurrence.**
**A** Overall survival, local progression free survival and regional progression free
survival curves for mice treated with a late neck resection depicted in Supple-
mentary Fig. 4E (Tumor+LN removal, $n = 6$; Tumor removal, $n = 6$; Sham surgery,
$n = 6$; and no surgery, $n = 4$). **B** Schematic of experimental design to look at rates of
regional progression in the setting of tumor only SBRT and anti-CD25 with or
without a distant tumor ($n = 10$ for buccal and flank tumor group and $n = 15$ for only
buccal tumor group). Created with BioRender.com. **C** Overall survival, local pro-
gression free survival, and regional progression free survival curves for mice
implanted with a buccal tumor or a buccal tumor and a flank tumor. 95% C.I. is
shown in dotted lines ($n = 25$ mice). **D** Regional progression free survival and cancer

free survival of mice treated with ENI, tumor only SBRT, or Sentinel Lymph node
SBRT (ENI, $n = 10$; tumor only SBRT, $n = 10$, and Sentinel LN SBRT, $n = 10$). Experi-
mental design is depicted in Supplementary Fig. 7F. **E** Tumor growth curves for
mice treated with or without sentinel lymph node removal ($n = 15$ for both groups).
To the right, the individual curves are shown. **F** Overall survival, local progression
free survival, and regional progression free survival curves are shown for mice
treated with sentinel lymph node removal ($n = 15$ for both groups). A log-rank
(Mantel−Cox) test was conducted to determine the survival difference between
treatment groups. Significance was determined if the $p$-value was <0.05\*, <0.01\*\*,
and <0.001\*\*\*. Source data are provided as a Source Data file. $p$-values are indicated
for figures **E**. \*\*\*0.0001, and **G**. \*\*0.0021, \*\*0.0018.

spread and risk of regional recurrence. The advent of immunotherapy,
however, has revolutionized interest in developing therapeutics aimed
at enabling the body's own immune system to generate durable cancer
responses. In that vein, this calls for a paradigm shift in how we treat
HNSCC. Draining lymphatics house the majority of the tumor specific
T cells, a site where DCs prime and expand antigen specific T cells, and
where central memory is generated[17]. DLNs are also critical for the
presence of an abscopal effect[30] and radiation of lymph nodes, more so
than circulating blood, has been shown to drive radiation induced
lymphopenia[31]. And yet current standard of care calls for compre-
hensive elective nodal targeting with surgery and/or radiation[6]. At a
time when almost all immunotherapy trials are failing in HNSCC, a
disease that is often treated with curative intent radiation, we must
question what the immunological consequences of ENI are on tumor
response, especially in the context of immunotherapy. Using pre-
clinical models of HNSCC including recently developed ones that
mimic the natural history of spread, we sought to understand the
outcomes of ENI on local, regional, and distant control with long
interval follow-up up to 200 days. We also conducted comprehensive
multicompartmental analysis to examine the immune effects within
the tumor microenvironment, as well as the nodal and blood com-
partments. We observed profound differences in both primary and in
distant tumor growth and metastasis when comparing mice treated
with ENI to mice treated with tumor only SBRT. The differences
observed between mice treated with ENI and tumor only SBRT was due
to systemic immune effects dependent on the primary tumor's DLNs.
This was further confirmed by using an upfront neck dissection,
another standard of care treatment for HNSCC, to show that the DLNs
are essential for a response to SBRT and immunotherapy. A model for
how we believe ENI is decreasing systemic immunity is provided in
Fig. 6K. Using data from mice and a canine clinical trial we also show
that ENI is associated with a systemic decrease in circulating T cells
which likely contributes to a dampened systemic immune response.

Our data suggest that ENI is driving the reduction in circulating
lymphocytes. This finding is corroborated by our data showing that ENI
is decreasing circulating T cells in mice, in canines, and as has been
reported in conventionally treated human patients with HNSCC[27,28].
Further supporting these findings are data showing no change in lym-
phocyte counts over time in a human trial of gross disease only treat-
ment with hypofractionated RT (SBRT), although a direct comparison of
patients treated with ENI versus tumor only irradiation would have to be
done in a clinical trial. Even when controlling for radiation field volume,
conventionally fractionated regimens where radiation is delivered in
repeated 1.8−2 Gy daily fractionation in 30 to 35 fractions, can result in
significant daily exposure of the blood volume to RT and lymphopenia as
has been demonstrated in the clinical literature[26–28]. However, momen-
tary exposure of the blood to RT in 3 to 5 hypofractionated sessions
(SBRT) separated by days would result in significantly less exposure of
the blood to RT. To be specific to the current work, the tumors in our
study were irradiated for 103 s while the lymph nodes were irradiated for
30 s. This is a minimal amount of time for arguing that lymphopenia is
primarily due to large blood volume exposure to RT as the blood is
getting minimal exposure to radiation as it travels through the radiation

field, especially compared to conventionally fractionated regimens.
Finally, our previous work has shown that treatment with Fingolimod
(FTY720), a treatment that traps differentiating effector T cells in sec-
ondary lymphoid organs, eliminated any benefit to radiation immu-
notherapy in our models[7]. This further supports that concept that the
draining lymph nodes are the site of action.

A key finding from this study is the essential role of CD4 T cells in
establishing and maintaining an anti-cancer immune response. It is well
known that CD8 T cells are essential for immune mediated tumor cell kill,
but more recently CD4 T cells have been recognized as critical players in
a variety of cancer models[32–36]. CD4 T cells are classically known as
supporting cells that maintain the immune response and support CD8 T
cell-mediated cell kill by providing survival signals like IL-2[37–39]. However,
it has been shown that antigen-mediated activation of both CD8 T cells
and CD4 T cells is indispensable for a robust anti-tumor immune
response[36,40]. MHC II binding antigens, which are presented only to CD4
T cells, were also recently used to vaccinate patients against cancer[41].
Vaccinating with MHC II binding antigens led to superior treatment
outcomes when compared to MHC I presented antigens[41]. Altogether
these findings underscore the importance of CD4 T cells and their ability
to maintain a long-term anti-tumor immune response. Here, we similarly
show the importance of CD4 and CD8 T cells in both treatment efficacy,
systemic immunity, and the adaptive immune response following
rechallenge. The importance of antigen-specific CD4 T cells in producing
a robust anti-tumor response that was identified in our studies stands to
inform future antigen-specific vaccination efforts.

Our data showed that the timing of surgery relative to radiation
and immunotherapy matters. When surgery was done after radiation
and immunotherapy, after systemic immunity has developed, it did
not affect local, distant, or regional recurrence. However, upfront
surgery followed by radiation immunotherapy was detrimental to local
tumor control and blunted an immune response. These data are con-
sistent with recent studies which have demonstrated that surgery
preceding immunotherapy did impair the development of systemic
memory response, compared to surgical resection occurring after
neoadjuvant immunotherapy[42]. These data are also consistent with
recent critical analysis of sentinel lymph node resection versus neck
dissection in patients with HNSCC[43]. The clinical implications for these
findings are substantial in that they suggest that neoadjuvant immu-
notherapy followed by delayed surgery that allows for sufficient time
for systemic immunity to develop, might be preferable to either
neoadjuvant immunotherapy followed by immediate surgery or
upfront surgery followed by immunotherapy.

Not only were mice treated with tumor only SBRT able to eradicate
local and distant tumors, but this treatment was also able to induce
epitope spreading. To overcome the genetic diversity of distant metas-
tasis, treatments aimed at enhancing epitope spreading will be key to
controlling systemic metastases that are significantly different from the
primary tumor[22,23,36,44,45]. In establishing metastasis, a cancer cell(s) needs
to drastically change gene expression to break away from the primary
tumor and establish a new growth in sometimes a completely new tissue
type[46]. So not only can metastases represent heterogenous group of
cells, but it often undergoes subclonal expansion and can become more

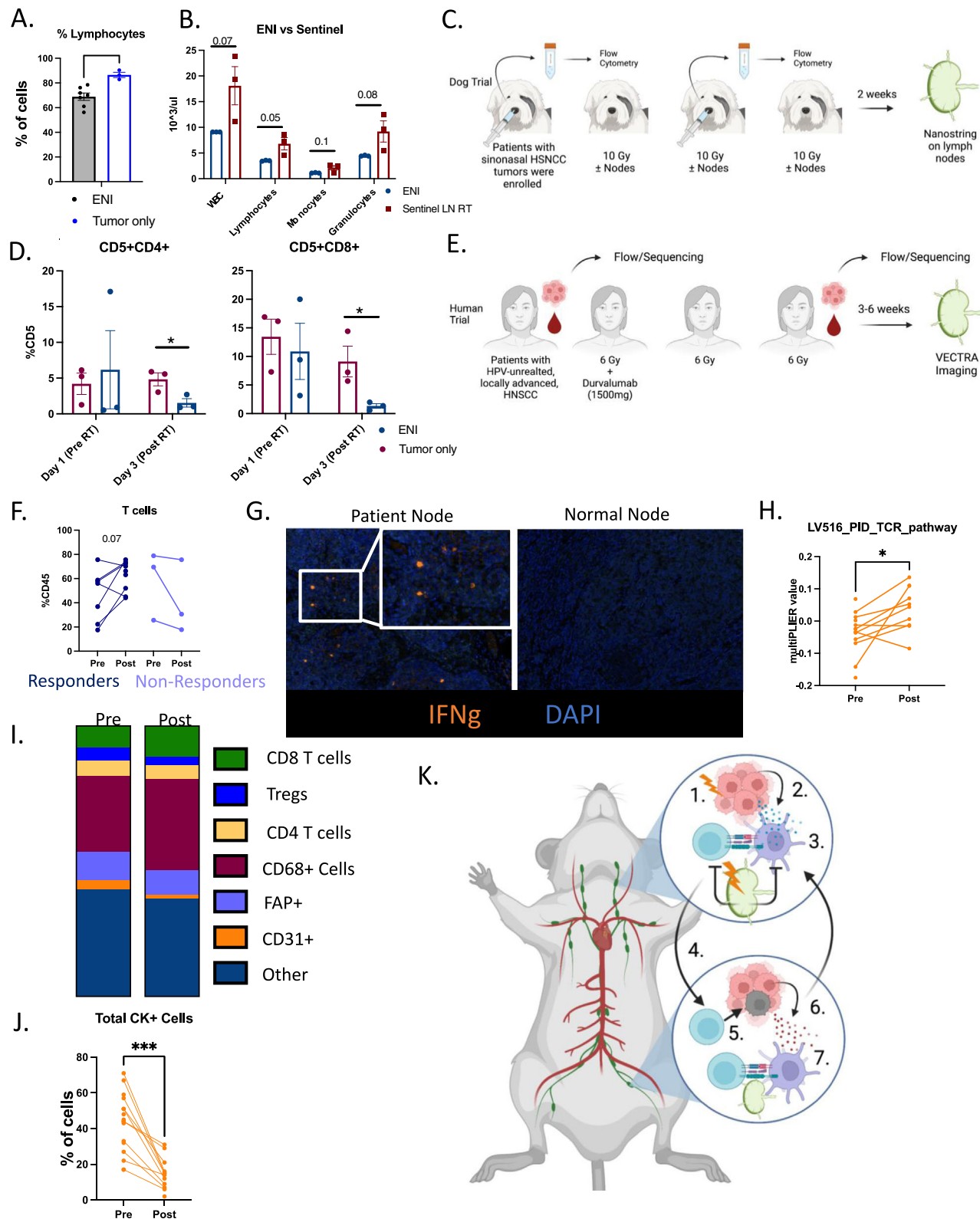

immunologically inert[47]. Our data show that treating a non-OVA primary tumor with tumor only SBRT was able to elicit a systemic OVA-specific response which was dampened by ENI. Although an increase in OVA-specific CD4 T cells in the blood could be a result of increased survival and not clonal expansion, an increase in OVA-specific activation in the DLNs suggests that the systemic increase is at least in part a result of clonal expansion. This was coupled with our findings that combination SBRT with immunotherapy can lead to eradication of distant metastasis

and generation of T effector memory cells in the blood. These data suggest that by avoiding ENI, and combining primary tumor only SBRT with immunotherapy, epitope spreading can be triggered to eliminate not only the primary tumor but also distant metastasis.

While a robust local and distant response was triggered with combination primary tumor only SBRT and immunotherapy, an unexpected finding was the persistence of regional metastasis. This may be due to the nature of lymphatic metastases versus

**Fig. 6 | Tumor only SBRT increases immune responses in canines and humans with HNSCC. A** Complete blood count from B16-OVA mice from Supplemental Fig. 2B (ENI, $n = 7$; tumor only, $n = 3$). **B** Complete blood count from LY2 mice from Fig. 5D (Sentinel LN, $n = 3$; ENI, $n = 3$). **C** Schematic of the clinical trial conducted in dogs diagnosed with sinonasal cancer and treated with SBRT with or without ENI. **D** Quantitation of CD4 (CD5+CD4+) and CD8 (CD5+CD8+) T cells from nasal lavages of dogs (ENI, $n = 3$; tumor only, $n = 3$). **E** Schematic of human phase I/Ib clinical trial using neoadjuvant SBRT and Durvalumab. **F** Quantitation of the percentage of CD45+ cells that are CD3+ T cells in the blood of patients before and after treatment with SBRT and durvalumab (Responders, $n = 9$; Non-responders, $n = 3$). **G** Representative multispectral fluorescent images of lymph nodes from patients and normal control nodes. IFNg is in orange. **H** MultiPLIER quantification of genes involved in TCR activation and genes downstream of TCR activation ($n = 14$ patients). **I** Quantification of cells within the TME of patients before and after treatment. **J** Quantification of cancer cells (CK+) in patients before and after

treatment ($n = 18$). **K** Model of how SBRT triggers a systemic immune response and how ENI reduces that response. 1. SBRT increases antigen in the TME, 2. antigen is acquired by DCs, which 3. present antigen to T cells in DLNs. These activated T cells act on the local tumor, but also 4. enter systemic circulation and go to the distant tumor site and 5. kills cancer cells which 6. release new antigen upon cell death. 7. These new antigens are presented to T cells in the distant DLN and triggers additional antigen-specific T cells to distant tumor only antigens. ENI decreases this systemic immune response by decreasing DC activation of antigen-specific T cells in the DLNs by radiation induced cell death. A two-tailed un-paired Student's $t$ test was used for mouse and canine data. A two-tailed paired Student's $t$ test was used for the human data. Significance was determined if the $p$-value was <0.05*, <0.01**, and <0.001***. The error bars represent the standard error of the mean (±SEM). Source data are provided as a Source Data file. $p$-values are indicated for figures **A**. **0.0071, **D**. CD4 *0.0375, CD8 *0.0451, **H**. *0.0346, and **J**. ***0.0002. Panels (**C**, **E** and **K**) created with BioRender.com.

non-lymphoid metastases. Lymph node metastases tend to be more polyclonal than distant metastases[22,45]. There is some evidence that cancer cells residing within lymph nodes have a decrease in high endothelial venules reducing/preventing the infiltration of tumor specific T cells to re-enter the DLNs[48]. Interestingly, we observed that regional metastasis started to grow noticeably only once both local and distant control was achieved.

A significant reduction in regional metastasis was observed with implementation of sentinel lymph node resection after primary tumor only SBRT and immunotherapy treatment has been delivered. This coincides with a time when DLNs are no longer essential for sustaining the systemic immune response, and there is circulating anti-tumor immune cell population. Sentinel lymph node resection was able to eliminate regional metastasis. These data are consistent with the documented importance of sentinel lymph node resection in both breast cancer and melanoma[49,50]. Treatment for breast cancer has evolved from radical mastectomy and bilateral axillary lymph nodes dissection to a sentinel lymph node biopsy (SLNB) and limited dissection[49] with ongoing trials examining the effects of omitting ENI (NCT03488693, NCT01872975). Similarly, melanoma treatment has migrated towards reducing the number of lymph nodes removed[50,51]. Ongoing clinical trials in HNSCC, such as HN006 (NCT04333537), examining the role of SLNB in early-stage oral cavity cancer, and other trials aimed at eliminating uninvolved neck irradiation[52], if successful, stands to pave the road for adopting this approach into HNSCC treatment guidelines. An alternative approach to surgical removal of sentinel lymph nodes would be sentinel lymph node irradiation which we showed was able to eliminate regional recurrence as well as surgical removal. This provides a non-invasive way of eliminating regional recurrence without permanently altering a patient's lymphatic circulation.

There are some limitations to these studies. First, the number of mice used in the surgical study (Supplemental Fig. 1E) was not sufficiently powered to determine a significant difference. The findings from this exact experiment, however, were recently replicated by another group independently[14]. In a similarly designed experiment, with a large sample size, Saddawi-Konefka et al. showed that early removal of draining lymph nodes decreases the effectiveness of immunotherapy[14]. Second, while it can be argued that the percent differences between the groups in the circulation could have been a result of lymphopenia, this would have likely affected all of the circulating leukocytes and not just ratios of specific cell types. Additionally, the decrease in percent CD8 T cells in the blood was corroborated by our findings in the primary and distant DLNs, as well as the primary and distant tumor. These data suggest that such population differences are not limited to the blood or due to lymphopenia. Third, a detailed kinetics study to further explore of how lymphocytes are trafficking between the blood, DLNs, and tumor after nodal irradiation over time

would be required to precisely determine the changes occurring in the nodes.

Finally, we used a hypofractionated RT regimen that is not considered standard of care in the definitive setting. This was done purposeful as gross tumor only hypofractionated RT is currently being tested in clinical trials in combination with immunotherapy[53,54] (NCT03635164, NCT05053737, NCT05085496, NCT03546582). Early results from these trials are indicative that hypofractionated RT can induce a larger immune response and enhance the effects of immunotherapy compared to conventional RT[4,15] (NCT03635164). As all clinical trials using immunotherapies combined with traditional RT have failed in HNSCC[4,55], we expect that these findings will be of high translational relevance in the setting of hypofractionated RT. Despite these limitations, our data strongly support future HNSCC clinical trial design with lymphatic sparing approaches and hypofractionated RT, particularly in the context of immunotherapy.

## Methods
### Cell Lines
LY2 (HNSCC), P029 (HNSCC), B16-OVA (melanoma), and 4T1 (breast cancer) were used for the in vivo studies. The Ly2 cell line was acquired from Dr. Nadarajah Vigneswaran (University of Texas Health Science Center, Houston, TX). P029 cell line was provided in collaboration with Xiao-Jing Wang at University of Colorado Anschutz Medical Campus, Department of Pathology. The B16-OVA cell lines was acquired from Dr. Rachel Friedman at the University of Colorado Anschutz Medical Campus, Department of Immunology. The 4T1 cell line was acqruired from Dr. Jill Slansky at the University of Colorado Anschutz Medical Campus, Department of Immunology. The HEK293-FT was provided in collaboration with Dr. Molishree Joshi at the University of Colorado Anschutz Medical Campus, Department of Pharmacology. Cell lines were cultured in appropriate media; LY2 and P029 cell lines were grown in DMEM-F12 with 10% FBS and 1% primocin/fungin. B16-OVA and 4T1 cell lines were grown in DMEM-F12 or RPMI with 10% FBS and 1% primocin/fungin, respectively.

P029 cell line was generated as follows: $K15\text{-}CrePR1$, $LSL\text{-}Kras^{G12D}$ and $Smad4^{f/f}$ mice on a C57BL/6J background were interbred and tail snips genotyped to establish tri-genic mice with Cre recombinase driven by the keratin15 promoter which activates expression of oncogenic mutant $Kras^{G12D}$ and deletes the $Smad4$ tumor suppressor in stratified epithelia as previously described[56]. Female mouse P029 developed a spontaneous skin lesion on the cervical region that was allowed to reach a diameter of 2 cm at which time the mouse was sacrificed, and tumor harvested for histological evaluation and cell line generation. The tumor was minced with scalpels, dissociated on a gentleMACS Tissue Dissociator using C tubes (Miltenyi) and incubated 40 min at 37 °C in 1 mg/mL Type II Collagenase (Worthington). Tissue suspensions were rinsed in PBS using centrifugation between washes

and initially cultured in complete media (DMEM/F12 media containing 10% FBS and 1x primocin antibiotics) for 7 days. To encourage epithelial cell growth and reduce fibroblast growth, cells were cultured in serum free keratinocyte media (Gibco) supplemented with 2 ng/mL EGF and 1x primocin. A stable, proliferating cell line, P029, was established after 2 weeks of culture and 4 passages to expand epithelial cells and eliminate fibroblasts. To verify tumor establishment and metastasis capabilities, P029 cell line was transplanted to the flanks of recipient female C57BL/6J mice using 50,000 cells in 50% matrigel/50% PBS (Corning) and monitored for 6 weeks when tumors reached 2 cm in diameter. Full necropsy and histological evaluation demonstrated P029 cells metastasize to the lung, liver, and lymph node.

To generate LY2-OVA cells: Five hundred thousand HEK293-FT cells were transfected with 2ug of pLVX-puro-cOVA-IRES-BFP (Addgene plasmid #135074) and 2ug of packaging viral mix (1:2 ratio of psPAX2 (Addgene plasmid #12260) and (Addgene plasmid #12259)) to generate lentiviral (LV) particles is a well of a six-well plate. Two ml of LV was collected 3 days post-transfection. 1 mL LV was used to transduce 500,000 LY2 cells. Media was changed ~24 h post-transduction. Transduced cells were selected with puromycin (1ug/mL) for 5–10 days.

## Mice

All mice were handled and euthanized consistent with the ethics guidelines and conditions set and overseen by the University of Colorado, Anschutz Medical Campus Animal Care and Use Committee. The study has been approved by the Institutional Animal Care and Use Committee (IACUC). Five- to six-week-old C57Bl/6 and DO11 female mice were obtained from the Jackson Laboratory (Bar Harbor, Maine, USA) and the BALB/c mice were obtained from Charles River Laboratories and were used for in vivo studies. Murine LY2 (BALB/c) and P029 (C57Bl/6) cell lines were implanted orthotopically into the buccal mucosa as previously described[57]. Murine cell lines B16 (C57Bl/6) and 4T1 (BALB/c) were implanted subcutaneously into the skin above the buccal muscle and into the mammary fat pad, respectively. Flank tumors were also implanted subcutaneously for all these models except for the P029 model. Mice were appropriately age matched and were randomized into groups, with treatment beginning when tumor volume was between 150–200 mm$^3$. Tumor measurements were conducted twice weekly using digital calipers and the tumor volume was calculated using the following equation, $V = (A \times B^2)/2$ mm$^3$, where $A$ and $B$ are the longer and shorter diameters of the tumor, respectively. For tumor studies the following cell numbers were implanted into the orthotopic location for each tumor cell line and twice as many cells were implanted into the flank for each model: LY2 $1 \times 10^6$ cells were implanted in the buccal and $2 \times 10^6$ in the flank per BALB/c mouse, for P029 $5 \times 10^4$ cells were implanted per C57/BL6 mouse, for B16 $1 \times 10^5$ cells were implanted per C57/BL6 mouse, and for 4T1 $1 \times 10^5$ cells were implanted per BALB/c mouse. Based on our approved animal protocol (Protocol# 00250), if the implanted buccal or mammary pad tumor exceeds 1000 mm$^3$ or a flank tumor exceeds 2000 mm$^3$ in a single plain measurement or if the tumors become ulcerated the mice were euthanized. Almost all our mice were euthanized before reaching these maximum tumor size limits. There were a few instances (Supplementary Fig. 2B) where the tumor limits were exceeded due to an unusually fast-growing tumor. We abided by our protocol by euthanizing these mice as soon as these numbers were exceeded.

## Canine samples

For the canine cancer study, all experimental protocols were reviewed and approved by the Colorado State University (CSU) Institutional Animal Care and Use Committee and the Clinical Review Board (IACUC #1058). Informed consent was obtained from all clients prior to enrollment of their dogs into the trial. Dogs with sinonasal cancer were randomized for treatment with SBRT (10 Gy × 3) targeted to primary

tumor+/− RLNs (n = 3 per treatment group). Tumor only irradiation was targeted to the sinonasal tumor (GTV + 2 mm PTV) without any nodal irradiation. Tumor + ENI, however, included the inonasal tumor (GTV + 2 mm PTV) plus bilateral submandibular and retropharyngeal LNs. The histological subtypes included 3 patients with carcinomas, 1 with sarcoma, and 2 malignant tumors that were characterized as aggressive histology consistent with neoplasia (e.g. carcinoma, atypical sarcoma). The histologies were distributed equally among the groups. The tumors were sampled via nasal lavage with warm saline using a validated technique for evaluating the immune profile of the canine nasal microenvironment. These cellular samples were collected pre-SBRT and following fractions 2 and 3 of SBRT. Two weeks post-SBRT, ipsilateral mandibular LNs were surgically extirpated for analyses.

Dogs were anesthetized and 10–20 mL (depending on the size of the dog) of pre-warmed sterile PBS solution was administered into the primarily affected nostril with a sterile, shortened red rubber catheter on the tip of the syringe, and the fluid backflow was collected from the nostrils in 50 mL conical tubes. This was repeated three times and the fluid containing the tumor-associated cells was pooled. Pooled samples were filtered through a 70 mm cell strainer to remove large debris and mucus. Samples were centrifuged and the pellets resuspended in PBS. Flow cytometry was performed with the nasal lavage samples at each time point to determine cell types. Cells were immunostained with the following fluorochrome-conjugated antibodies: T cells: CD5-PE (Clone: YKIX322.3 eBiosciences, 12-5050-42); CD4-PB (Clone: YKIX302.9 Bio-Rad, MCA1038PB), and CD8-APC (Clone: YCATE55.9 eBiosciences, 17-5080-42). Flow cytometric analysis was performed using a Beckman Coulter Gallios flow cytometer and data will be analyzed using FlowJo software.

RNA was extracted from frozen tumor tissues using the RNeasy Plus Mini kit (QIAGEN) following manufacturer protocol. Depending on starting material, samples were eluted in 30–50 µL RNase-Free water. Samples were initially checked for quantity and purity on a Nanodrop ND-1000 Spectrophotometer (Thermo Fisher) prior to being stored at −80 °C until further processing. Samples were additionally quantity and quality checked using the RNA High Sensitivity assays on the Qubit 2.0 Fluorometer (Invitrogen/LifeTechnologies) and 5200 Fragment Analyzer Automated CE System (Agilent), respectively. NanoString gene expression analysis was performed using the Canine IO 360 panel. Nanostring analysis was performed with the nCounter Analysis FLEX system at the University of Arizona Genetics Core. Gene expression count data was analyzed via nSolver software.

## Human samples

The trial (NCT03635164) was carried out in accordance with Good Clinical Practice (GCP) as required by applicable United States (US) laws and applications, including but not limited to United States (US) Code of Federal Regulations (CFR) applicable to clinical studies (45 CFR Part 46, 21 CFR Part 50, 21 CFR Part 56, 21 CFR Part 312, and/or 21 CFR Part 812). We confirm that relevant regulations regarding the use of human study participants and was conducted in accordance with the criteria set by the Declaration of Helsinki. Dr. Karam assures that no changes to the protocol took place without documented approval from the Institutional Review Board (IRB). All personnel involved in the conduct of this study have completed Human Subjects Protection Training. Written informed consent and HIPAA authorization was obtained from the patient prior to performing any protocol-related procedures, including screening evaluations. Clinical outcomes for this trial have been published, please see Darragh et al, Nature Cancer, In Press[58].

The Human Immune Monitoring Shared Resource (HIMSR core) at the University of Colorado School of Medicine performed the immunostaining of patient tumor and DLN tissue using the Perkin

Elmer Vectra 3 instrument. The same protocol was followed as previously described[59]. Color images were processed with inForm software version 2.5 and 2.6. Quantification was done in Akoya Phenoptoreports in R version 4.1.0 and 4.1.1, including cell percentages, cell densities, phenotyping, and spatial analysis. We used MultiPLIER to analyze our RNA sequencing data for cell type population level data. The code for MultiPLIER is publicly available at https://github.com/greenelab/multi-plier from Taroni et al.[60].

### Radiation Design

Radiation was delivered using an X-RAD SmART irradiator (Precision X-ray, Madison CT) with a beam energy of 225 kVp to mice under isoflurane anesthesia. Treatment planning and Monte Carlo simulations were performed using SmART-ATP software (SmART Scientific Solutions, Maastricht, the Netherlands, v.2.0.20200916). Radiation doses were 24 Gy in three fractions for tumor and 15 Gy in three fractions for ENI.

For HNSCC models, buccal tumors were treated using a tangent beam positioned to not intersect a pair of opposed lateral beams used for nodal irradiation. For the 4T1 breast cancer model, mice were placed in a lateral recumbent position with the tumor gently pulled away from the chest using a plastic ring. A 1 cm circular beam was then aligned using fluoroscopy to irradiate the tumor while hitting minimal normal tissue. Mice receiving nodal irradiation were then repositioned under a custom collimator designed to produce four 1 cm circular beams targeting the inguinal and axillary lymph nodes. Dosimetry for this collimator was confirmed using Gafchromic EBT3 film (Ashland Global, Wilmington DE).

For sentinel lymph node irradiation, mice were irradiated with a fractionated dose of 8 Gy × 3 using the XRAD SmART Irradiator (Precision X-ray, Madison CT). Irradiation of the primary tumor and sentinel lymph nodes was done by angling the beam to 325 degrees and positioning it such that the tumor and lymph node were irradiated while avoiding as much normal tissue as possible, as well as other lymph nodes. The presence of the sentinel node in the irradiated field was confirmed in fluoroscopy using 2.0 mm metal pellets (Beekley Medical, Bristol CT) placed at the approximate position of the node.

### Surgical Design

The mice were anesthetized with isoflurane at 5% (2 L/min oxygen flow) and maintained at 1.5 to 3% isoflurane (adjusted to maintain adequate breathing and prevent response to a toe pinch). Eye ointment was used to prevent corneal dryness and blindness. The neck and right facial skin were shaved with an electric shaver, then cleaned with 70% ethanol. An incision (about 1–2 cm) was made in the neck and face area using scissors. In a group of mice, the buccal tumor and bilateral superficial cervical lymph nodes were visualized and then removed. In other subgroups, the removal of buccal tumor only, bilateral superficial cervical lymph nodes only, and ipsilateral superficial cervical draining lymph nodes was performed. Sham surgery included making the skin incision with blunt dissection of neck and right facial soft tissues, without removal of tissue. Pressure with a sterile cotton tip applicator was used to control bleeding. Mice's respirations, heart rate, and tactile temperature were monitored during the surgeries. Either skin glue or absorbable suture was used to close the incision. The mice were awakened on a warm circulating water heat pad and observed during recovery. Mice were given 100 μL (0.1 mg/kg) of buprenorphine HCL for pain management for two days post-operatively. Mice were monitored for poor wound healing, dehiscence, bleeding, hematoma, seroma, infection, pain, poor oral intake, and dehydration.

### Anti-CD25 and depletion antibodies

αCD25 was provided in collaboration with Roche Pharmaceuticals. αCD25 was given at a concentration of 3 mg/kg. αCD25 was administered weekly via I.P. injection beginning one day prior to the beginning of RT. For studies not utilizing radiation therapy, I.P. injections were administered after tumor implantation, at a time point equivalent to one day before RT. αCD4 and αCD8 was administered twice weekly at 10 mg/kg via I.P. injections.

### Adoptive transfer of DO11 T cells

Spleens and lymph nodes were harvested from DO11 mice. Spleens and lymph nodes were collected in ice cold HBSS and then filtered through a 70um nylon cell strainer to produce a single cell suspension. Spleen, but not LNs, samples were centrifuged at 400$g$ for 5 min and resuspended in RBC lysis buffer (Invitrogen), using HBSS to neutralize the lysis buffer. CD4 T cells were isolated using EasySep Mouse CD4 T cell Isolation Kit following the kit instructions (Stemcell Technologies). After isolation, the cells were counted and resuspended in DPBS. $1 \times 10^5$ CD4 T cells were adoptively transferred into mice via tail vein injection.

### Flow cytometry

Tumor, blood, and tumor draining lymph nodes were harvested and processed for flow cytometric analysis. Tumor tissue was chopped and incubated in Collagenase III (Worthington) for 30 min at 37 °C. After incubation, tissue was passed through a 70um nylon cell strainer to produce a single cell suspension. Blood was collected via cheek bleed or intracardiac puncture. After centrifugation, red blood cells were lysed using RBC lysis buffer (Invitrogen), using HBSS to neutralize the lysis buffer. Lymph nodes were similarly processed by mechanical separation into single cell suspension. Blood was immediately centrifuged after collection and resuspended in RBC lysis buffer as described above. Cells were transferred into 24 well plates and incubated with monensin and brefeldin to prevent release of cytokines, and stimulated with PMA/ionomycin cocktail for 4 h at 37 °C. Following incubation, cells were incubated in FC block (CD16/CD32 antibody, Tonbo bioscience) for 15 min. Cells were then incubated in Live/Dead Fixable Aqua Viability Stain Kit (Invitrogen) in the dark for 20 min. Cells were then stained for surface markers and incubated for 30 min. For analysis of immune cells, the following antibodies were used at the dilution recommended by the manufacturer: Percp-CD45 (Clone: 30-F11 BD Biosciences, 557235), BUV805-CD3 (Clone: 17A2 BD Biosciences, 741982), BUV496-CD4 (Clone: GK1.5 BD Biosciences, 612952), BB515-CD8 (Clone: 53-6.7 BD Biosciences, 564422), PE-Dazzle/594-TIM-3 (Clone:B8.2C12 BiLegend, 134014), FITC-Granzyme B (Clone: QA16A02 BioLegend, 371106), Percp-eF710-OX-40 (Clone: OX-86 eBioscience, 46-1341-82), eF450-CXCR4 (Clone: 2B11 eBioscience, 48-9991-80), BV421-Tbet (Clone: 4B10 BioLegend, 644815), BV570-CD44 (Clone:IM7 BioLegend, 103037), APC-IL-2 (Clone: JES6-5H4 eBioscience, 17-7021-82), PE-Cy7-NKp46 (Clone:29A1.4 BioLegend, 137618), BV605-DNAM1 (Clone:TX42.1 BioLegend, 133613), BUV737-IFNg (Clone: XMG1.2 BD Biosciences, 612769), Alexa Fluor 532-Foxp3 (Clone: FJK-16s eBioscience, 58-5773-82), BV786-CD25 (Clone: 3C7, BD Biosciences, 564368), APC-eF780-Ki67 (Clone: SolA15 eBioscience, 47-5698-82), BV650-MHCII (Clone: M5/114.15.2 BioLegend, 107641), PE-Cy5-CD11c (Clone N418 BioLegend, 117316), BUV615-PD-1 (Clone: J43 BD Biosciences, 752299), BV750-TNFa (Clone: MP6-XT22 BioLegend, 506308), PE-TCF7/1 (Clone: S33-966 BD Biosciences, 564217), BV711-CD103 (Clone: 2E7 BioLegend, 121435), Percp-Cy5.5-CCR7 (Clone: 4B12 BioLegend, 120116), BUV395-CD18 (Clone:C71/16 BD Biosciences, 740225), Alexa Fluor 647-Cleaved Caspase-3 (Clone: C92-605 BD Biosciences, 560626), SB436-CD39 (Clone: 24DMS1 eBiosciences, 62-0391-80), SB436-CD69 (Clone: H1.2F3 ThermoFischer, 14-0691-82), APC-CD62L (Clone: MEL-14 BioLegend, 104412), BV480-CD27 (Clone: LG.3A10 BD Biosciences, 746742), PE-TCR DO11.10 (Clone: KJ1-26 BioLegend, 118508), Percp-Cy5.5-CD80 (Clone: 16-10A1 BD Biosciences, 560526), PE-Cyanine7-CXCR3 (Clone: 173 eBioscience, 25-1831-82) and Percp-eF710-EOMES (Clone: Dan11mag eBioscience, 46-4875-82).

After surface staining, cells were fixed and permeabilized using the Foxp3 perm/fix kit (Invitrogen) overnight. Following incubation,

cells were stained for intracellular markers and incubated for 30 min. Samples were then run on a Cytek Aurora spectral cytometer at the University of Colorado Diabetes Research Center Flow Cytometry Core. Fluorescence minus one controls were used to determine gating strategy. Flowjo analysis software and Cytobank was used for data analysis. Data was initially analyzed in Flowjo, and CD45 + cell populations were uploaded into Cytobank for clustering analysis. T-SNE clustering was conducted and that was followed by FlowSOM population determination.

**Mouse multispectral immunofluorescent staining and analysis**
The Human Immune Monitoring Shared Resource (HIMSR core) at the University of Colorado School of Medicine performed the immunostaining of tumor tissue using the Perkin Elmer Vectra 3 instrument. The same protocol was followed as previously described[59]. Color images were processed with inForm software version 2.6.

**Statistical analysis**
All statistical analyses were processed using GraphPad Prism v9 or SAS 9.4 (SAS Institute Inc 2013). Quantification of H&E stains were done in ImageJ (Fiji 1.0). The mean differences between the two groups were determined by two-tailed, unpaired student's t-tests. To compare the mean differences, we used one- or two-way analysis of variance (ANOVA) with Tukey's post hoc test for multiple comparisons. The Chi-squared test with continuity correction or Fisher's exact test (if any cell count <5) was used to test the proportion difference between groups. Time-to-death by tumor-related symptoms was plotted using Kaplan–Meier (KM) curves and the survival difference between groups was compared using log-rank (Mantel–Cox) tests. Statistical significance was set at $p < 0.05$. Multiple testing adjustment with Benjamini–Hochberg procedure was further performed to control for false-positive rates (FDR). The significance is denoted by asterisks, $*p < 0.05$, $**p < 0.01$, $***p < 0.001$, and $****p < 0.0001$. All data are reported with mean ± SEM (standard error or the mean).

**Reporting summary**
Further information on research design is available in the Nature Portfolio Reporting Summary linked to this article.

## Data availability
The canine nanostring data has been deposited in the GEO database under the accession code GSE217028. The clinical trial protocol is available online at clinicaltrials.gov with the following clinical trial number: NCT03635164. The human RNA sequencing data has been deposited in the GEO database under the accession code GSE210287 which includes individual de-identified data of the patients sequenced and is publicly available. The remaining data are available within the Article, Supplementary Information or Source Data file. Source data are provided with this paper.

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

## Acknowledgements

The authors would like to thank the small irradiator core at the University of Colorado Anschutz for help in designing and implementing the ENI protocols. We would like to thank The University of Colorado diabetes Research Center, specifically the flow cytometry core, which is funded by NIDDK grant #P30-DK116073. We would like to thank Xiao-Jing Wang's lab for providing us with the P029 HNSCC cell line. We would also like to thank Rachel Friedman for the B16-OVA cell line and we would like to thank Jill Slansky for the 4T1 cell line. We would also like to acknowledge our funding sources: R01 DE028529-01 (SDK), R01 DE028282-01 (SDK), 1P50CA261605-01 (SDK, XJW), DE028420 (XJW), VA I01 BX003232 (XJW), VA IK6BX005962 (XJW), CCTSI Colorado Pilot Grant Award (SDK & KB), and National Institutes of Health grant F31 DE029997 (LBD).

## Author contributions

S.D.K. conceptualized and designed the study; contributed to the analysis, writing, and review of the manuscript. L.B.D. was involved in the conceptual framework and contributed to the experimental design, conducted experiments, data collection, analysis, and writing of the manuscript. J.G. contributed to experimental design, conducted experiments, data collection, data analysis, and review of the manuscript. T.P. conducted experiments, data collection, data analysis, and review of the manuscript. V.S., B.C., B.N., D.N., M.K., N.A.O., S.C., and M.H. were involved in data collection and/or analysis, and review of the manuscript. M.J. generated the LY2-OVA cell line and reviewed of the manuscript. K.N generated the PO29 cell line and reviewed the manuscript. X.J.W. and R.K. shared their lab resources, provided conceptual feedback, and contributed to the manuscript review process. Y.Z. contributed to the statistical analysis. M.K.B ran the canine clinical trial, provided conceptual feedback, and contributed to the manuscript review process. M.A. and S.D. provided conceptual feedback and contributed to the manuscript review process.

## Competing interests

Dr. Karam receives clinical funding from Genentech and Ionis that does not relate to this work. She receives clinical trial funding from AstraZeneca, a part of which is included in this manuscript. She also receives preclinical research funding from Roche for work related to the anti-CD25 antibody, which is utilized as immunotherapy in this manuscript. The remaining authors declare no competing interests.
