## [Peer Review File · Nature Communications]

REVIEWER COMMENTS

Reviewer #1 (Remarks to the Author): with expertise in cancer radiotherapy/immunotherapy

The manuscript by Darragh and colleagues addresses an important clinical question that has implications for the treatment of patients with local therapies, such as radiation and surgery, in the setting of cancer immunotherapy. Elective nodal irradiation has been previously shown by Marciscano et al. (Clinical Cancer Res 2018) to reduce the response to combinations of radiation and immune checkpoint inhibitors in mice. From an immunological perspective it is known that healthy (i.e., not metastatic) lymph nodes are required for optimal priming of anti-tumor immune responses. However, in the clinic the need to balance standard-of-care procedures that have proven benefits (e.g., reducing regional recurrence) with new immunotherapies has led to the design of trials that failed to elicit therapeutically significant anti-tumor immune responses. The work described here aims at gathering preclinical evidence to support new trial designs, particularly for H&N cancer.

Strengths of the manuscript are the orthotopic mouse H&N cancer models, the testing of both radiation and surgical removal of nodes, and the use of canine patients. However, the quality of the experimental work suffers from a general lack of scientific rigor: control groups, number of mice used, and statistical analyses are missing in most experiments shown. Analyses of the immune cells are descriptive but lack direct evaluation of tumor-specific T cell responses. Figure legends often contain insufficient details to understand the data shown. The effects of ENI on the irradiated nodes is not properly evaluated: is cellularity decreased, temporarily or persistently, and is the architecture altered? Some immune cell subsets are evaluated, but percentages of a given cell subset are not providing the full picture of how the node function is affected by radiation. There are no controls for the effects of large field radiation (such as used to irradiate bilateral inguinal and axillary nodes, or the neck which is likely to include the carotid artery in the field): lymphopenia is only evaluated in Figure 6, in one model, and there is no discussion about whether it could be a major confounding factor. Overall, many of the conclusions of the authors are speculative and not well-supported by the data. Additional concerns are detailed below.

1) The immunotherapy used in combination with RT is an antibody against CD25, based on a prior study that it was effective at inducing tumor regression in the H&N cancer models used. Surprisingly, Tregs are never evaluated in the current study to confirm that they are depleted. In addition, in the prior work a single 10 Gy RT dose was used with anti-CD25. Are Tregs also increased when using RT at 8GyX3, and are they equally important in the B16-OVA and 4T1 models used in the current study?

2) Figure 1B: is the difference between RT and RT+ENI statistically significant?

Figure 1 D and E: it is not clear why the lung metastases are not quantified at euthanasia by histology, which provides a more direct evidence than microCT. A control arm of anti-CD25 alone and RT alone would help determine the contribution to the response of the irradiation of the primary tumor, at the doses and fractionation used in this study, which is different from previously published experiments.

3) Figure 1F and G: B16-OVA and 4T1 are also treated with anti-CD25, but it is not clear if this therapy combination works in these models. The supplementary figure legend mentions that for B16 and 4T1 RT doses are given 4-5 days apart. Why this choice? In addition, these mice get a lot of radiation since bilateral axillary and inguinal nodes are irradiated, and a serious lymphopenia is not ruled out.

4) Figure 1H: what does it mean tumor removal? If the tumor was surgically removed why the curve is the same as sham surgery and no surgery? There is no significant statistical difference between the groups.

5) Figure 1I: the chosen markers are peculiar and do not include any evaluation of Tregs, which are the targets of the immunotherapy. Are the markers reported the only ones that happened to be significantly different between the two groups among a larger panel?

6) Figure 2: OVA is a strong antigen and priming of endogenous T cells can be easily detected using tetramers and/or ex vivo stimulation with OVA peptides. Why the adoptive transfer is needed?

7) Figure 2C is difficult to interpret and the legend provides no explanation. The text of results is also unclear "Similarly, using the non-antigen specific LY2 cell line, we observed a decrease in distant tumor growth in the LY2-OVA cell line" but the figure shows no such a decrease. What are the authors trying to prove?

8) Figure 2F and H: the indirect evaluation of T cell priming in the irradiated and non-irradiated nodes by looking only at LFA1 and CD44 expression is a surprising choice, since this is a central hypothesis of the study. A decrease in DC is reported, but a major concern is that the data are expressed as percentage, without any information about LN cellularity: are irradiated nodes smaller? The results presented are superficial and not compelling at all. Despite the conclusion that "ENI dampens an antigen-specific immune response both locally and distantly", such antigen-specific response is never measured.

9) Figure 3B: the interpretation of the results is largely speculative, since all data are presented as percentages, and no specific hypothesis is really tested: for example, the increase in circulating cD11c+ MHC-II + cells is interpreted as pDC without actually testing pDC markers. In addition, it is unclear if any of the cured mice comes from ENI group.

10) Figure 3N: the legend is insufficient. It is not too surprising that removal of LN at day 19 after priming has already occurred is not having an impact. However, this important point should be tested. For example, tumor-specific T cells are likely to be expanded in the spleen and other secondary lymphoid organs, as well as the buccal mucosa, something that can be tested easily using the OVA model. Again, the group labelled "tumor removal" makes no sense.

11) Figure 4: The hypothesis that the response of the flank tumor is due to epitope spreading after irradiation of the buccal tumor cannot be tested using the experimental design described. Only the flank tumor expresses OVA, and mice are given OVA-specific T cells, which are activated in the flank tumor and its draining nodes. A contribution from T cells primed against other antigens expressed in the irradiated buccal tumor is not tested and not required for the response. The slightly lower response of mice treated with ENI (not clear if statistically significant) may reflect other immunosuppressive effects of the large field that is irradiated in the neck: a control of a similar field that does not include draining nodes should be performed. In fact, the results of this experiment put into question other conclusions about what ENI is doing in terms of affecting tumor-specific T cell priming.

12) Figure 5A: mice that are followed for recurrence are 4 mice w/o ENI (from Figure 1A and B) and 2 mice with ENI. These numbers are too low to make any conclusions. Figure 5D: there is no mention of the number of mice evaluated that have or not two tumors and have or not cleared completely the original tumor after RT+anti-CD25, even in the supplementary figure this information is not available. The poor rigor of this work makes it very hard to follow and interpret these results. Figure 5 F and G do not mention number of mice and statistical significance. In addition, if only one "sentinel" lymph node is removed, it is rather strange that the tumor does not recur in other local nodes. Was the removed sentinel node involved by microscopic metastasis?

13) Figure 6A: lymphopenia is evaluated but not clear why in the B16 model used only once. 6B: in the dogs lymphocytes are quantified in nasal lavage, so this cannot be used to evaluate lymphopenia (which refers to blood). It appears that only 3 animals were evaluated in each group.

Reviewer #2 (Remarks to the Author): with expertise in lymphatics, cancer immunology/immunotherapy

Summary: Here, the authors utilize models of HNSCC to investigate the impacts of tumor-alone (lymph node-sparing) radiation therapy compared to radiation therapy targeting both the tumor and lymph node on responses to immunotherapy. They conclude that irradiating the tumor-draining lymph nodes impairs immunotherapy responses. Likewise, dissection of tumor-draining lymph nodes is shown to impair immunotherapy responses. The authors investigate the immunological ramifications of these interventions and show altered T cell and dendritic cell dynamics. The work is then extended using a canine model, with similar findings. The overall goal of the project is significant, and some of the results are compelling; however, there are several issues with the study that should be addressed:

Major:

1. The use of anti-CD25 mAb for immunotherapy is atypical as compared to anti-PD1 or anti-CTLA4. Can the authors contextualize this in more typical immunotherapeutic regimens? Do these results extend to more common treatment modalities?
2. Very low numbers of animals are utilized in the flow cytometry experiments, decreasing the reliability of this data. n higher than 3 would enhance the impacts of this work.
3. The authors identify several differences in factors involved in cell migration (Lfa-1 and CCR7). Besides the use of these as identification markers for cellular phenotype, are their potential ramifications on cellular migration? This would be of relevance because migration can affect the number of circulating cells and cell infiltration into the tumor microenvironment(s).
4. The authors show that OVA-expressing tumors are less sensitive to the impacts of ENI (in both the B16-OVA and LY2-OVA model). What are the ramifications of this on the conclusions of the work? Does tumor antigenicity alter the need for lymph node sparing?
5. The gating strategies used are not consistent with what is generally accepted in the field. Quad gates throughout the paper would be more rigorous. This is of particular importance in Figure 2I, where the improper gating changes the conclusion of the figure – a quad gate at $\sim 4 \times 10^4$ on the x-axis and $\sim 2 \times 10^4$ on the y-axis would be much more reasonable, and shows very different results. The conclusions from 2I should thus be re-examined.

Minor:

6. There are a number of sentences which are grammatically confusing – a thorough readthrough and editing would benefit the readability of the manuscript in general.
7. Plasmacytoid dendritic cells are defined within this work by MHCII expression; however, conventional dendritic cells can express MHCII as well (pDCs are typically defined by B220 expression in flow). The authors should amend this to simply identify an MHCII+ DC subset, which does not change the conclusions of the paper.
8. Statistics are missing in Figure 4B, but statistical significance is claimed in the text. Please clarify this. Statistics are also missing throughout Figure 5 and would be useful.
9. Timepoints are missing in Figure 5B.
10. More details of the effects of radiation on the lymph node would be helpful. Did the irradiation affect the lymphatic vessels or lymph transport? Or was the effect restricted to immunological events in the node itself?

Reviewer #3 (Remarks to the Author): with expertise in HNSCC, cancer immunology/immunotherapy

This is an elegant pre-clinical study using multiple orthotopic murine models to show that elective nodal irradiation, compared with radiation only to the primary tumor, dampens multiple facets of the anti-tumor immune response including immunologic memory and long-term tumor control. Regional tumor control was further enhanced by performing a delayed sentinel node biopsy or neck dissection. These experiments are highly relevant to head and neck cancer, which is commonly treated with definitive radiation. The manuscript is well written and innovative, in keeping with this high-impact

journal.

Major concerns:

- 1) Figure 6, showing systemic lymphopenia after RT, seems tangential to the rest of the paper. Consider removing it or finding a way to tie it in better. A schema/graphical abstract of the take-home points from all experiments might be more useful.
- 2) The Discussion should have a paragraph acknowledging limitations of the study. For example, small numbers of animals in the surgical experiments.
- 3) SBRT with high doses of 8 Gy is highly immunogenic, but this is not how head and neck cancer is typically treated in the previously-untreated setting. Thus, it is unclear if these results can be extrapolated to the more standard, hyperfractionated regimen. This deserves mention as another potential limitation of the study.
- 4) Use of anti-CD25 as a form of immunotherapy along with RT is casually mentioned. Better description of the rationale for using it (to enhance immune effects of RT? Is it better than anti-PD-1?) is warranted.
- 5) The Discussion also needs to acknowledge other studies in the literature showing that the DLN is critical for abscopal responses to RT (e.g., Buchwald et al., JTC 2020).

Minor concerns:

- 1) The one sentence summary should be more concise.
- 2) The use of the term "HPV-unrelated HNSCC" is confusing and a bit distracting in the abstract, since HPV is not a focus of this study, and oral cancer is not typically HPV related anyway. Consider leaving it in the introduction but taking that out of the abstract.
- 3) Consider adding more info to the figure panels, e.g., adding "buccal" and "flank" labels to 1B, labels for the mouse model/cell line used in more of the panels, etc. to make the figures easier to interpret without referring to the legends each time.
- 4) Figure 3M does not appear to be mentioned anywhere in the Results text.

Reviewer #4 (Remarks to the Author): with expertise in veterinary radiation oncology

Dear authors, I would like to compliment you on a fine manuscript. I was asked to provide feedback within my specific expertise, veterinary radiation oncology.

General point: (this might already be stated elsewhere in the submission files): please make sure to mention ethics approval for procedures in dogs.

Your findings might be of future relevance for ENI treatment for certain diseases also in dogs. The reduction of antigen presenting and T cell homing genes and increase of genes associated with immunosuppression seem caused by the radiation-induced partial elimination of the specific lymphocyte population. If this is necessarily of consequence in tumor response of sinonasal tumors and other tumors as well remains to be investigated in detail. It is, however, of potential high interest specifically in tumors where the immune system's effect is thought to be of high relevance and / or where tumors are already treated concurrently with immunotherapy, such as canine malignant oral melanoma.

Specific points which would help me as a reader to better follow your findings:

M&M

I recommend specifying if all the 6 dogs included were diagnosed with "squamous cell carcinoma" or to adapt the wording accordingly (for example "sinonasal carcinoma").

Lines 610: please specify which lymph nodes were considered «regional lymph nodes» for treatment? E.g., were only the ipsilateral mandibular RLN treated with RT or others treated as well and just the ipsilateral excised? I would assume not only the mandibular, but also medial retropharyngeal lymph nodes to be draining the nasal cavity for most sinonasal tumors and most of sinonasal tumors

occurring as bilateral disease, this could be of interest to the reader.

Lines 610ff: tumor sampling for diagnostic purposes via nasal lavage is a somewhat described technique, however, I do not see, how such a lavage can be trusted to represent the tumors immune cell population. Did you investigate this prior or is there any literature you could quote on this technique?

REVIEWER COMMENTS

Reviewer #1 (Remarks to the Author): with expertise in cancer radiotherapy/immunotherapy

The manuscript by Darragh and colleagues addresses an important clinical question that has implications for the treatment of patients with local therapies, such as radiation and surgery, in the setting of cancer immunotherapy. Elective nodal irradiation has been previously shown by Marciscano et al. (Clinical Cancer Res 2018) to reduce the response to combinations of radiation and immune checkpoint inhibitors in mice. From an immunological perspective it is known that healthy (i.e., not metastatic) lymph nodes are required for optimal priming of anti-tumor immune responses. However, in the clinic the need to balance standard-of-care procedures that have proven benefits (e.g., reducing regional recurrence) with new immunotherapies has led to the design of trials that failed to elicit therapeutically significant anti-tumor immune responses. The work described here aims at gathering preclinical evidence to support new trial designs, particularly for H&N cancer.

We thank the reviewer for recognizing the importance of the clinical implications of our work outlined in this manuscript. We would like to point out that unlike the Marciscano et al. paper that used a flank model of melanoma [1], which was cited in this work, the implantations, and the irradiation, here are done, orthotopically, in the head and neck region or the mammary fat pad. Also different is the fact that we utilize, head and neck cell lines, melanoma, and breast cancer cell lines. We also use metastatic models to examine distant tumor growth and systemic immunity in addition to local tumor growth.

Strengths of the manuscript are the orthotopic mouse H&N cancer models, the testing of both radiation and surgical removal of nodes, and the use of canine patients.

We thank the reviewer for highlighting that this paper uses orthotopic models of HNSCC and the use of both radiation and surgical models to evaluate the importance of the DLNs in response to immunotherapy. We would like to note that we also used an orthotopic model of breast cancer (4T1) and a cutaneous model of melanoma as well (B16-OVA). We have also added human correlative data from a recently completed Phase I/IB clinical trial (Figure 6) to support our findings throughout the manuscript. The manuscript for this Phase I/IB trial is provided as a related manuscript file in the re-submission.

However, the quality of the experimental work suffers from a general lack of scientific rigor: control groups, number of mice used, and statistical analyses are missing in most experiments shown.

We thank the reviewer for bringing these details to our attention. This reviewer mentions a lack of an RT only group for Figure 1B. This group was not originally included as all our past research using this model, in 5 different published research articles including numerous animal experiments, has shown that RT alone does not eradicate local or distant tumors [2-6]. However, to address the reviewer's comment we have repeated that experiment and the data are now included in the updated manuscript (Figure 1B).

We have also carefully re-examined the manuscript and ensured that the number of mice used per experiment is included in each figure legend and that every graph includes statistical analysis. To address the reviewer's point, statistical significance is now also included in the text and highlighted in the figures.

Analyses of the immune cells are descriptive but lack direct evaluation of tumor-specific T cell responses.

We would like to thank the reviewer for bringing this point of view to our attention. Reporting percentages of populations is generally how immune data is presented in high impact journals [7-10] and is not considered descriptive in this space. We also included counts of cells when relevant in the manuscript (see Figure 4G) as well as additional information on tumor-specific T cells (See Figure 4D and H).

Figure legends often contain insufficient details to understand the data shown.

We have now included additional details in the legends which we hope satisfies the reviewer's concerns.

The effects of ENI on the irradiated nodes is not properly evaluated: is cellularity decreased, temporarily or persistently, and is the architecture altered? Some immune cell subsets are evaluated, but percentages of a given cell subset are not providing the full picture of how the node function is affected by radiation.

We would like to thank the reviewer for being interested in these additional facets of our experimental outcomes. We did evaluate immune cell populations relevant for immune cell priming and activation in the lymph nodes (Figure 1I). We also evaluate cell death in the lymph nodes, which is likely the most direct way that RT is affecting lymphocytes in the DLNs (Supplemental Figure 5C). To address the reviewer's concerns on cellularity, this is now evaluated in the DLNs and no difference was observed between the groups between the two groups (Supplemental Figure 2C). The literature supports that large volume irradiation decreases lymphocytes systemically and persistently [11, 12]. To corroborate this concept, we had included lymphocyte count data from our mice and intranasal lavage data from our canine model that showed a decrease in lymphocytes in B16-OVA ENI treated animals (Figure 6A). To further validate these findings, we have now included evaluation of lymphocytes in our LY2 mouse model (Figure 6B and Supplemental Figure 7B). Again, subsets of immune cells by percentages are a common and accepted way of evaluating immune cells [7-10] and is not considered descriptive especially when we use T cell depletion antibodies to manipulate these populations. Please see a more detailed response above.

There are no controls for the effects of large field radiation (such as used to irradiate bilateral inguinal and axillary nodes, or the neck which is likely to include the carotid artery in the field): lymphopenia is only evaluated in Figure 6, in one model, and there is no discussion about whether it could be a major confounding factor. Overall, many of the conclusions of the authors are speculative and not well-supported by the data. Additional concerns are detailed below.

We agree that large field irradiation will affect development of lymphopenia as has been demonstrated with conventional radiation regimens in human patients [11]. The question of whether exposing circulating lymphocytes to radiation, versus nodal basins, as the instigating origin of radiation-induced lymphopenia is a great one that is indeed deserving of discussion. ENI and more specifically, the number of lymph nodes in the treatment volume, is likely the driving factor behind reduced circulating and intratumoral lymphocytes in our studies for the following reasons. Controlling for time of radiation treatment and frequency of delivering the radiation as confounders, systemic lymphopenia from hypofractionated radiation (SBRT) should yield decreased lymphopenia, due to blood volume irradiation, compared to conventional fractionation regimens. Even when controlling for field volume, conventionally fractionated regimens where radiation is delivered in repeated 1.8-2 Gy daily fractionation in 30 to 35 fractions, can result in significant daily exposure of the blood volume to RT and lymphopenia as has been demonstrated in the clinical literature [13-15]. However, momentary exposure of the blood to RT in 3 to 5 hypofractionated sessions (SBRT) separated by days would result in significantly less exposure of the blood to RT. To be specific to the current work, the tumors in our study were irradiated for 103 seconds while the lymph nodes were irradiated for 30 seconds. This is a minimal amount of time for large blood volume exposure to RT, especially compared to standard of care. To lend evidence to the argument that ENI is the driver of RT related decreases in lymphocytes, we have now also included data limiting radiation to just the sentinel lymph node, which shows increased circulating lymphocytes compared to mice treated with ENI (Figure 6B). We also show that ENI is specifically reducing antigen-specific T cells in the DLNs via apoptosis (Supplemental Figure 5C). The reduction in lymphocytes occurs only after treatment and only in the ENI treated group. Finally, our previous work has shown that treatment with Fingolimod (FTY720), a treatment that traps differentiating effector T cells in secondary lymphoid organs, eliminated any benefit to radiation immunotherapy in our models [3]. This further supports that concept that the draining lymph nodes are the site of action. These concepts are now included in the Discussion section of the revised manuscript.

The immunotherapy used in combination with RT is an antibody against CD25, based on a prior study that it was effective at inducing tumor regression in the H&N cancer models used. Surprisingly, Tregs are never evaluated in the current study to confirm that they are depleted.

In previous work we have repeatedly demonstrated that anti-CD25 depletes Tregs in this model [2-4]. We have added that information into the text. Additionally, proof of Treg depletion with the anti-CD25 for our initial experiment is now provided in Supplementary Figure 1B.

In addition, in the prior work a single 10 Gy RT dose was used with anti-CD25. Are Tregs also increased when using RT at 8GyX3?

We thank the reviewer for reviewing our past work and how it relates to this current manuscript. In this manuscript we show that Treg depletion with RT at 8Gy x 3 induces tumor eradication and that this only happens when anti-CD25 is combined with RT. Based on that alone one can safely conclude that Tregs are a key player in resistance to therapy. However, to address the reviewer's comment, we conducted an experiment to show that 8Gy x 3 increases Tregs in the TME to a similar level as our historical experiments using 10Gy x 1 [2]. Please see below. This information

has been added to the text and we included our new data showing the levels of Tregs in the TME after 8Gy x3 in Supplemental Figure 1B, with a reference to our historical (published) data and for the reviewer the data is shown as a head-to-head comparison below.

And are they equally important in the B16-OVA and 4T1 models used in the current study?

The dependence of the B16 OVA on Tregs has long been demonstrated [16-18] even in the context of radiation [19]. Similarly, the 4T1 model is dependent on Tregs especially in the context of RT [20, 21]. It is important to note, that in our study, neither the B16 nor the 4T1 were treated with a-CD25 to illustrate the point that ENI versus tumor-only RT drives the difference in response in these radioresistant models.

2) Figure 1B: is the difference between RT and RT+ENI statistically significant?

Compared to our “no RT” control, only the “tumor-only RT” group had a significant decrease in the percentage of mice that had flank tumors at the end of the study as determined by an Fischer’s Exact test. This has now been included.

Figure 1 D and E: it is not clear why the lung metastases are not quantified at euthanasia by histology, which provides a more direct evidence than microCT.

We chose to use microCT to evaluate the mice at multiple timepoints. To use histology would require euthanasia of the animals at the same date. Using the microCT we were able to evaluate mice, over time, while still gathering tumor volume data. To evaluate our samples at time of death would be to introduce another variable, time to death. Additionally, unless one takes serial sections of an entire lung, of every lung sample collected, and conducts 3D re-construction of that, using histology at time of death would not allow for proper evaluation of the entire lung for metastasis like microCT does. We were fortunate to have access to high a resolution microCT at our institution, and for research rigor, included microCT images for increased rigor or metastasis evaluation. To confirm that the metastases observed on microCT were indeed metastases we evaluated H&Es and confirmed that the metastases observed on microCT are indeed metastases based on H&E (Supplementary Figure 1I). We have clarified in the text that we are only concluding that there is a delay in lung metastasis in tumor-only treated mice as evaluated by microCT.

A control arm of anti-CD25 alone and RT alone would help determine the contribution to the response of the irradiation of the primary tumor, at the doses and fractionation used in this study, which is different from previously published experiments.

Thank you for bringing this to our attention. We had included a control arm for anti-CD25 alone in the first experiment and an RT alone arm in the breast cancer model experiment, but an RT control arm has now also been included in Figure 1B. Similar to our previous publications [2-6], no eradication was observed in these mice.

3) Figure 1F and G: B16-OVA and 4T1 are also treated with anti-CD25, but it is not clear if this therapy combination works in these models.

These two models were used as additional models to underscore the differences between ENI and tumor-only RT. Anti-CD25 was not used in these models for exactly the reason the reviewer states.

The supplementary figure legend mentions that for B16 and 4T1 RT doses are given 4-5 days apart. Why this choice?

To mimic clinical delivery with hypofractionated radiation. For an 8Gy x 3 hypofractionated regimen delivered clinically, on trial or off trial, it is common practice to give it twice a week, over a period of two weeks especially concurrent with immunotherapy.

In addition, these mice get a lot of radiation since bilateral axillary and inguinal nodes are irradiated, and a serious lymphopenia is not ruled out.

That is precisely the point: that ENI results in a decrease in circulating lymphocytes. The experiments were conducted in the best possible way to limit the field in small animals and still treat all of the draining lymph nodes electively. The entire objective of the experiment was to show that nodal irradiation results in a reduction in circulating lymphocytes. Additionally, this experiment merely corroborates our experiments in three other models where the irradiation field was much smaller, suggesting that it is not the size of the irradiation field but more the lymph nodes that are being irradiated that is driving the effect. The size of irradiation, blood volume exposed, and the effect of irradiation on lymphopenia are thoroughly addressed in a previous comment.

4) Figure 1H: what does it mean tumor removal? If the tumor was surgically removed why the curve is the same as sham surgery and no surgery? There is no significant statistical difference between the groups.

We apologize for the confusion. Tumor resection here is an R2 resection or tumor debulking. These tumors are highly aggressive and without RT and a-CD25, there is rapid regrowth. We reported that the p-value was 0.07 and showed a trend in tumor growth increase in the mice that had DLNs removed prior to therapy. The impact of this experiment is highlighted in a recent paper [22].

5) Figure 1I: the chosen markers are peculiar and do not include any evaluation of Tregs, which are the targets of the immunotherapy. Are the markers reported the only ones that happened to be significantly different between the two groups among a larger panel?

Tregs were not included in the paper as they are depleted in both groups, as we have shown in previous publications [2] and in this manuscript (Supplementary Figure 1B). As the goal of this experiment was to examine differences in cell populations between ENI and tumor-only treated mice, we concluded that evaluating a cell population that is depleted in both groups would not provide any insight into the differences in systemic immunity between the two groups. As we and many others have shown that anti-CD25 depletes Tregs [2, 23, 24], examination of activation markers is more central to the research question at hand, especially in terms of targets of immunotherapy. This immunotherapy also results in activation of effector T cells [2, 3, 23, 24]. We had included only the relevant information within the manuscript as these were the markers that we found significant in Figures 2-4 as well. However, indeed numerous other markers were included in our flow cytometry panel. As this experiment was done using cheek bleeds, only major populations were interpretable. But, to address the reviewer's concern, all these data are now included in Supplemental Figure 1J and 1K.

6) Figure 2: OVA is a strong antigen and priming of endogenous T cells can be easily detected using tetramers and/or ex vivo stimulation with OVA peptides. Why the adoptive transfer is needed?

Balb/c, unfortunately, do not have an equivalent OTI model. As such, we had to search for CD4 T cells that have TCRs specific for MHCII. MHCII tetramers are known to have a weaker binding affinity to their targets than MHCI tetramers [25]. In lieu of tetramer staining, we therefore used adoptive transfer that recognized DO11 cells with the KJ antibody to examine antigen specificity. This is a commonly used method and has been used extensively in the literature [26-29].

7) Figure 2C is difficult to interpret and the legend provides no explanation. The text of results is also unclear "Similarly, using the non-antigen specific LY2 cell line, we observed a decrease in distant tumor growth in the LY2-OVA cell line" but the figure shows no such a decrease. What are the authors trying to prove?

We apologize that the experimental design was unclear. The experimental design for Figure 2C was provided in Figure 2B and we hope that that is now very clear from the text and the figure legend. More detailed figure legends have now been included as requested. Statistics have been added. The two groups are different as illustrated in the tumor growth curves and as was demonstrated in Figure 1B which was the experiment we used to determine [30] differences as the mice were able to live until tumor resolution.

This is a great question and one that we spent a lot of time considering. We apologize that it was not clear in the text. We have added wording to the manuscript to make our key point clearer. The key point here is that ENI, consistently, and independent of tumor model, increases distant tumor growth, a point of high translational relevance. The effect on local tumor growth is different between OVA and WT LY2 models. Local and distant tumor growth differences are shown in Figure 2C. It was an interesting, but not surprising, finding that in OVA LY2 models, which have a strong exogenous antigen, there is also a difference in primary tumor growth compared to our WT LY2 models. This is what we wanted to highlight. Regardless of antigenicity though, we found that ENI is detrimental in all models.

8) Figure 2F and H: the indirect evaluation of T cell priming in the irradiated and non-irradiated nodes by looking only at LFA1 and CD44 expression is a surprising choice, since this is a central hypothesis of the study. A decrease in DC is reported, but a major concern is that the data are expressed as percentage, without any information about LN cellularity: are irradiated nodes smaller? The results presented are superficial and not compelling at all. Despite the conclusion that “ENI dampens an antigen-specific immune response both locally and distantly”, such antigen-specific response is never measured.

We thank the reviewer for bringing this to our attention. CD44 and LFA-1 positive cells have been demonstrated to be the most strongly activated T cells due to DC priming and are considered antigen-experienced cells [26, 31-34]. We also disagree that percentages are not standard in the field. Here are just a few examples of high impact publications demonstrating the point: [7-10]. Histological analysis of the nodes showed no difference in cellularity, with similar amounts of cells in the DLNs of mice treated with ENI and Tumor only SBRT (Supplementary Figure 2C). Antigen specific cells were measured in the blood. However, an additional experiment was performed showing increased antigen specific cells in the TME shown in Figure 4D and H. We hope that we can convince the reviewer that these are not superficial findings. The compelling nature of the findings is driven by the strength of the evidence: This is demonstrated in four separate models. ENI reduces the effectiveness of therapy across all models. Differences in immune cell activation and effector T cells are shown in three different compartments: TME, Nodal, Blood.

9) Figure 3B: the interpretation of the results is largely speculative, since all data are presented as percentages, and no specific hypothesis is really tested: for example, the increase in circulating cD11c+ MHC-II + cells is interpreted as pDC without actually testing pDC markers. In addition, it is unclear if any of the cured mice comes from ENI group.

We apologize for the confusion. We did not use ENI treated mice in any of the cured analysis. This has been clarified. As we have addressed previously percentages are not considered speculative and it is how highly reputable immunology journals expect reporting as addressed in the previous comment. We have changed the reference of plasmacytoid DCs to just DCs. We would like to highlight that using CD4 and CD8 depletion models allowed us to confirm that these cells are driving memory response in Figure 3, which is direct evidence of the role of these cells in memory.

10) Figure 3N: the legend is insufficient. It is not too surprising that removal of LN at day 19 after priming has already occurred is not having an impact. However, this important point should be tested. For example, tumor-specific T cells are likely to be expanded in the spleen and other secondary lymphoid organs, as well as the buccal mucosa, something that can be tested easily using the OVA model. Again, the group labelled “tumor removal” makes no sense.

We apologize that the legend was considered insufficient. We have included additional information to the figure legend describing the experiment depicted in Figure 3N, which is shown

in a diagram in Supplemental Figure 3E and also described in the methods section. We agree that it is not surprising that removing the DLNs at Day 19 does not impact the primary tumor growth. However, the experiment was testing if the memory was housed in the DLNs or if it was systemic. As we implanted flank tumors after DLN removal, we can safely say that DLNs are not the only place where memory is maintained/stored.

As mentioned in a previous comment, the tumor removal was a debulking. The tumors grow back aggressively after surgery. The tumor removal surgery was and is extensively discussed in the methods section.

11) Figure 4: The hypothesis that the response of the flank tumor is due to epitope spreading after irradiation of the buccal tumor cannot be tested using the experimental design described. Only the flank tumor expresses OVA, and mice are given OVA-specific T cells, which are activated in the flank tumor and its draining nodes. A contribution from T cells primed against other antigens expressed in the irradiated buccal tumor is not tested and not required for the response. The slightly lower response of mice treated with ENI (not clear if statistically significant) may reflect other immunosuppressive effects of the large field that is irradiated in the neck: a control of a similar field that does not include draining nodes should be performed. In fact, the results of this experiment put into question other conclusions about what ENI is doing in terms of affecting tumor-specific T cell priming.

We apologize if the experimental design and purpose was unclear, but we respectfully disagree that the design of the experiment does not test this question. We are only treating the primary tumor with RT so only those antigens should be presented to the DCs initially. Epitope spreading is then evaluated by the observed increase in OVA T cell priming as the OVA antigen is not present in the primary tumor. It is experimentally unrealistic to do the large field proposal, in a mouse, without hitting lymph nodes, be it submandibular, interfacial, retropharyngeal, jugular digastric etc.

We have now included tumor antigen-specific T cell information in the manuscript showing that in the distant lymph node (the inguinal lymph node) there is an increase in Th1 (Tbet+) antigen-specific T cells and that antigen-specific CD4 T cells are increased in the flank tumor as well (Figure 4D and H).

12) Figure 5A: mice that are followed for recurrence are 4 mice w/o ENI (from Figure 1A and B) and 2 mice with ENI. These numbers are too low to make any conclusions. Figure 5D: there is no mention of the number of mice evaluated that have or not two tumors and have or not cleared completely the original tumor after RT+anti-CD25, even in the supplementary figure this information is not available. The poor rigor of this work makes it very hard to follow and interpret these results. Figure 5 F and G do not mention number of mice and statistical significance. In addition, if only one “sentinel” lymph node is removed, it is rather strange that the tumor does not recur in other local nodes. Was the removed sentinel node involved by microscopic metastasis?

The number is low because those are the only mice that survived from 10 initially. We agree that it is too small of a number to make any robust conclusions. We initially powered up this experiment

to examine regional recurrence just in the mice treated with tumor-only RT, as enough survived to make any reasonable conclusion. What we found was that having only a buccal tumor or having a buccal and a flank tumor did not impact the rate of regional metastasis. Based on these findings we were able to repeat the initial experiment that was underpowered with only buccal tumors so that more of the ENI mice would survive to observe regional metastasis. Using this experimental design, we found that 4/8 tumor-only treated mice developed regional recurrence while 0/9 ENI treated mice did. This data powers up our original findings showing that ENI eliminates regional recurrence in our model.

To address this question even further, we conducted a new experiment looking at gross tumor + sentinel lymph node irradiation compared to ENI or tumor-only RT. In this experiment we found that sentinel lymph node irradiation was able to reduce regional metastasis just as well as sentinel lymph node resection via surgery. 0/9 mice treated with sentinel lymph node irradiation developed regional metastases. This data has now been included in Figure 5.

Finally, to address whether or not sentinel nodes have micro metastatic disease, some SLNs of tumor only mice do have evidence of micrometastatic disease after treatment, we have now included that in Supplemental Figure 6D. From our experiments using sentinel lymph node removal in 15 mice and sentinel node irradiation in another 10 mice, we do not observe nodal metastasis in non-sentinel nodes in this model. Since we do not observe regional recurrence in all mice treated with tumor-only irradiation, we can postulate that there is micro metastatic disease in ~50% of mice and that this is eliminated with sentinel lymph node removal or irradiation. This is supported by our histological analysis of SLNs mentioned above.

Regarding the rigor comment, with all due respect, these are experiments that included 10 animals per group (buccal and flank arm) and 15 (for buccal only arm), totaling 25 animals. For figures 5F and G, 15 mice were included for each group, totaling 30 mice. The follow-up times for these experiments was 150 days or longer. Further, we would like to remind the reviewer that these results were repeated and validated across multiple tumor models, a testament to the rigor and validity of the findings.

13) Figure 6A: lymphopenia is evaluated but not clear why in the B16 model used only once. 6B: in the dogs lymphocytes are quantified in nasal lavage, so this cannot be used to evaluate lymphopenia (which refers to blood). It appears that only 3 animals were evaluated in each group.

We have included new circulating lymphocyte data for the LY2 model in Figure 6. The canine trial is a prospective design, so we were limited to 3 patients in each arm of the clinical trial. Data are also now included from a human trial with gross disease-only treatment. Please see the related manuscript files for more information on the human trial samples.

Reviewer #2 (Remarks to the Author): with expertise in lymphatics, cancer immunology/immunotherapy

Summary: Here, the authors utilize models of HNSCC to investigate the impacts of tumor-alone (lymph node-sparing) radiation therapy compared to radiation therapy targeting both the tumor

and lymph node on responses to immunotherapy. They conclude that irradiating the tumor-draining lymph nodes impairs immunotherapy responses. Likewise, dissection of tumor-draining lymph nodes is shown to impair immunotherapy responses. The authors investigate the immunological ramifications of these interventions and show altered T cell and dendritic cell dynamics. The work is then extended using a canine model, with similar findings. The overall goal of the project is significant, and some of the results are compelling; however, there are several issues with the study that should be addressed:

We would like to thank the reviewer for appreciating the potential impact of these findings and we hope that we have addressed their comments thoroughly.

Major:

1. The use of anti-CD25 mAb for immunotherapy is atypical as compared to anti-PD1 or anti-CTLA4. Can the authors contextualize this in more typical immunotherapeutic regimens? Do these results extend to more common treatment modalities?

We thank the reviewer for their insightful comment. Prior publications by our lab have demonstrated that aCTLA-4, with or without RT, failed to deliver any therapeutic benefit in the LY2 model [2]. Similarly, aPD-L1 with RT elicited a modest response in LY2 and MOC2 tumors, although in the end this amounted to nothing more than tumor growth delay [5]. aCD25 therapy, in conjunction with RT, however, outshined these more “typical” immunotherapeutic regimens leading to complete eradication of LY2 tumors and significant delay in the MOC2 model. These data served as a prelude and backbone for the experimental design in this manuscript, and due the superior therapeutic effect of aCD25, we chose not to investigate these other modalities. However, based on the effect seen in B16 and 4T1 models, we would expect the effects of ENI to extend to these other treatment modalities.

2. Very low numbers of animals are utilized in the flow cytometry experiments, decreasing the reliability of this data. n higher than 3 would enhance the impacts of this work.

We agree that having an n greater than three would increase the impact of this work. We have now done that by replicating our work in Figure 3 for the cured mouse group and have included that data in Supplemental Figure 3C. Throughout the manuscript we have noted in the legends our mouse numbers to show an increase in n.

We have also included in a paragraph in the discussion that we are limited in our conclusions of our upfront surgical experiment as there was low power. Another group has conducted this experiment in other models of HNSCC independently of our experiments and has shown that early removal of DLNs is detrimental for the efficacy of immunotherapies [22].

3. The authors identify several differences in factors involved in cell migration (Lfa-1 and CCR7). Besides the use of these as identification markers for cellular phenotype, are their potential ramifications on cellular migration? This would be of relevance because migration can affect the number of circulating cells and cell infiltration into the tumor microenvironment(s).

We agree that this is an important variable to test. To address this question, we examined CXCR3 and CCR7 expression by CD4 and CD8 T cells in the DLNs. We found that tumor only treated mice had an increase in CD4 T cells expressing CCR7, a marker that retains CD4 T cells in the LN. We also observed that the tumor-only treated mice had an increase in CD8 T cells expressing CXCR3, which is known to promote egress of T cells out of the lymph node and, in this case, most likely promote migration to the TME [35]. These data have been added to Supplemental Figure 5A and B. Additionally, since DLNs are known to have the most tumor antigen specific T cells, we anticipate that RT is decreasing the repertoire of T cells capable of recognizing tumor antigens. This would reduce the amount of T cells activated and homed to the TME, as validated by the observed decrease in antigen-specific T cells in the distant tumor (Figure 4D and H).

4. The authors show that OVA-expressing tumors are less sensitive to the impacts of ENI (in both the B16-OVA and LY2-OVA model). What are the ramifications of this on the conclusions of the work? Does tumor antigenicity alter the need for lymph node sparing?

Superb point and one to which we have given significant thought. The key finding here though is that, regardless of antigenicity, there are negative effects of ENI. In essence, antigenicity does not need to be evaluated before treatment is initiated on a patient as all will benefit from lymph node sparing therapies. This would have enormous translational implications as current trials are being designed to test this question. These points are now emphasized in the manuscript the Results section summarizing Figure 2. A cartoon has also been added to underscore this important point and summarize the findings of this manuscript in Figure 6K.

5. The gating strategies used are not consistent with what is generally accepted in the field. Quad gates throughout the paper would be more rigorous. This is of particular importance in Figure 2I, where the improper gating changes the conclusion of the figure – a quad gate at $\sim 4 \times 10^4$ on the x-axis and $\sim 2 \times 10^4$ on the y-axis would be much more reasonable, and shows very different results. The conclusions from 2I should thus be re-examined.

We thank the reviewer for their comment on our gating strategy. While we agree that the use of quadrant gating for conventional cytometry is generally a more appreciated approach, the use of this tactic for spectral cytometers is less reliable as the separation between positive and negative values is not as rigid, thus creating less distinction between such populations. Non-quadrant gating has been demonstrated in being just as robust as conventional quadrant gating at identifying unique immune subsets using spectral flow cytometers [36] and is the most commonly utilized in high impact journals such as nature cancer [37] and nature immunology [38]. We would like to direct the reviewer to our gating strategy in Supplementary Figure 3B where a visible population is observed for CD44+LFA-1+ cells. We did not adjust the gating strategy per sample as that would not be representative of the same population of cells being compared between samples.

Minor:

6. There are a number of sentences which are grammatically confusing – a thorough readthrough and editing would benefit the readability of the manuscript in general.

Our apologies. A more careful readthrough has now been performed.

7. Plasmacytoid dendritic cells are defined within this work by MHCII expression; however, conventional dendritic cells can express MHCII as well (pDCs are typically defined by B220 expression in flow). The authors should amend this to simply identify an MHCII+ DC subset, which does not change the conclusions of the paper.

We agree that the definition for pDCs could be stronger in the paper. We have now changed this to an MHCII+ DC population as suggested.

8. Statistics are missing in Figure 4B but statistical significance is claimed in the text. Please clarify this. Statistics are also missing throughout Figure 5 and would be useful.

We apologize for the oversight. Statistics have been added to Figure 4B and Figure 5.

9. Timepoints are missing in Figure 5B.

This has been added.

10. More details of the effects of radiation on the lymph node would be helpful. Did the irradiation affect the lymphatic vessels or lymph transport? Or was the effect restricted to immunological events in the node itself?

We agree that this is a very important question. We are working on exploring this answer in a subsequent manuscript as we believe that this requires substantial characterization.

Reviewer #3 (Remarks to the Author): with expertise in HNSCC, cancer immunology/immunotherapy

This is an elegant pre-clinical study using multiple orthotopic murine models to show that elective nodal irradiation, compared with radiation only to the primary tumor, dampens multiple facets of the anti-tumor immune response including immunologic memory and long-term tumor control. Regional tumor control was further enhanced by performing a delayed sentinel node biopsy or neck dissection. These experiments are highly relevant to head and neck cancer, which is commonly treated with definitive radiation. The manuscript is well written and innovative, in keeping with this high-impact journal.

We would like to thank the reviewer for appreciating the potential impact of this work and the models that were used to conduct this work.

Major concerns:

1) Figure 6, showing systemic lymphopenia after RT, seems tangential to the rest of the paper. Consider removing it or finding a way to tie it in better. A schema/graphical abstract of the take-home points from all experiments might be more useful.

Thank you for this feedback. We have now included additional data on systemic decreases in circulating lymphocytes from another mouse model (LY2) and from human patient data. Information on the human trial samples is provided in the related manuscript files. Excellent suggestion on including a visual abstract. This is now incorporated into Figure 6K to summarize the main takeaways from this paper.

2) The Discussion should have a paragraph acknowledging limitations of the study. For example, small numbers of animals in the surgical experiments.

Agreed. Done.

3) SBRT with high doses of 8 Gy is highly immunogenic, but this is not how head and neck cancer is typically treated in the previously-untreated setting. Thus, it is unclear if these results can be extrapolated to the more standard, hyperfractionated regimen. This deserves mention as another potential limitation of the study.

We do agree that this is currently not standard of care for HNSCC and have added it as a limitation of this study in the discussion section. We used hypofractionated RT as HNSCC clinical trials are now integrating hypofractionation into the previously untreated setting [39], NCT03635164, NCT05053737, NCT05085496, NCT03546582. Our lab currently has a Phase I and Phase II trial using hypofractionated radiation and immunotherapy. With promising results, we anticipate that standard of care will hopefully one day change to hypofractionated RT in the previously untreated setting when immunotherapy is on board. This study is now referenced in the text.

4) Use of anti-CD25 as a form of immunotherapy along with RT is casually mentioned. Better description of the rationale for using it (to enhance immune effects of RT? Is it better than anti-PD-1?) is warranted.

We agree with the reviewer that there was a lack of rationale behind the use of anti-CD25 in combination with RT. The introduction has been updated to include a description justifying its use in these experiments. Our past studies have shown that anti-CD25 is superior to anti-PD1 in our LY2 model of HNSCC [2, 23, 24].

5) The Discussion also needs to acknowledge other studies in the literature showing that the DLN is critical for abscopal responses to RT (e.g., Buchwald et al., JITC 2020).

Done.

Minor concerns:

1) The one sentence summary should be more concise.

We have shortened the one sentence summary.

2) The use of the term “HPV-unrelated HNSCC” is confusing and a bit distracting in the abstract, since HPV is not a focus of this study, and oral cancer is not typically HPV related anyway. Consider leaving it in the introduction but taking that out of the abstract.

Good point. We can refer to it as oral cavity cancer in the abstract.

3) Consider adding more info to the figure panels, e.g., adding “buccal” and “flank” labels to 1B, labels for the mouse model/cell line used in more of the panels, etc. to make the figures easier to interpret without referring to the legends each time.

Thank you for this feedback. Those labels have now been included and we hope it is now easier to interpret the figure.

4) Figure 3M does not appear to be mentioned anywhere in the Results text.
Our apologies for the oversight. This is now referenced.

Reviewer #4 (Remarks to the Author): with expertise in veterinary radiation oncology

Dear authors, I would like to compliment you on a fine manuscript. I was asked to provide feedback within my specific expertise, veterinary radiation oncology.

General point: (this might already be stated elsewhere in the submission files): please make sure to mention ethics approval for procedures in dogs.

Thank you for noticing that we had not included a statement on ethics approval. The following has been added to the manuscript in the methods section:

“For the canine cancer study, all experimental protocols were reviewed and approved by the Colorado State University (CSU) Institutional Animal Care and Use Committee and the Clinical Review Board (IACUC #1058). Informed consent was obtained from all clients prior to enrollment of their dogs into the trial.”

Your findings might be of future relevance for ENI treatment for certain diseases also in dogs. The reduction of antigen presenting and T cell homing genes and increase of genes associated with immunosuppression seem caused by the radiation-induced partial elimination of the specific lymphocyte population. If this is necessarily of consequence in tumor response of sinonasal tumors and other tumors as well remains to be investigated in detail. It is, however, of potential high interest specifically in tumors where the immune system’s effect is thought to be of high relevance and / or where tumors are already treated concurrently with immunotherapy, such as canine malignant oral melanoma.

Thank you for sharing your feedback. We appreciate the reviewer’s insight and recognition of our work especially as it relates to the canine population. Indeed, canine malignant melanoma is an important translational application of this work, and we have ongoing trials in this space. Melanoma was not included as we wanted to limit the trial sinonasal cancers, so the radiation treatment can be similarly applied and there would be no variation in nasal lavage sampling. As such melanoma was not included, since in the head and neck region, it mostly occurs with the oral cavity. This is currently being validated in a larger prospective trial. However, this trial included which speaks to the validity of the findings across different histological subtypes.

Specific points which would help me as a reader to better follow your findings:
M&M

I recommend specifying if all the 6 dogs included were diagnosed with “squamous cell carcinoma” or to adapt the wording accordingly (for example “sinonasal carcinoma”).

The dogs included in this study were affected with nasal tumors of various histotypes. We have adapted the wording to “sinonasal cancer”. However, a list of all the histologies is now included in the methods section. The histological subtypes included 3 patients with carcinomas, 1 with

sarcoma, and 2 with malignant tumors that were characterized as aggressive upon histological assessment and consistent with neoplasia (e.g. carcinoma, atypical sarcoma). The histologies were distributed equally among the groups.

Lines 610: please specify which lymph nodes were considered «regional lymph nodes» for treatment? E.g., were only the ipsilateral mandibular RLN treated with RT or others treated as well and just the ipsilateral excised? I would assume not only the mandibular, but also medial retropharyngeal lymph nodes to be draining the nasal cavity for most sinonasal tumors and most of sinonasal tumors occurring as bilateral disease, this could be of interest to the reader.

Thank you for your request for additional information. The following has been added:

The regional lymph nodes treated in the “elective nodal irradiation” canine study included bilateral mandibular and medial retropharyngeal lymph nodes. For further clarification, the Tumor only irradiation was targeted to the sinonasal tumor (GTV+ 2mm PTV_ without any nodal irradiation. Tumor + ENI, however, included the sinonasal tumor (GTV+ 2mm PTV) plus bilateral submandibular and retropharyngeal LNs.

Lines 610ff: tumor sampling for diagnostic purposes via nasal lavage is a somewhat described technique, however, I do not see, how such a lavage can be trusted to represent the tumors immune cell population. Did you investigate this prior or is there any literature you could quote on this technique?

Thank you for bringing this up for additional clarification. We have published on the efficacy of the nasal lavage technique to evaluate the immune profile of the canine nasal microenvironment using purpose-bred research dogs [40]. The nasal lavage technique for sampling and evaluating the local immune response within the nasal cavity of dogs affected with sinonasal tumors is a new approach developed within our group. We are working to understand correlations between nasal lavage immune cell populations and sinonasal tumor microenvironmental biopsy samples. In this study, however, the shifts in immune cell populations associated with ENI for dogs with sinonasal tumors represent the local immune effects within the canine sinonasal microenvironment, not the sinonasal tumor microenvironment.

Citations

1. Marciscano, A.E., et al., *Elective Nodal Irradiation Attenuates the Combinatorial Efficacy of Stereotactic Radiation Therapy and Immunotherapy*. Clin Cancer Res, 2018. **24**(20): p. 5058-5071.
2. Oweida, A.J., et al., *STAT3 Modulation of Regulatory T Cells in Response to Radiation Therapy in Head and Neck Cancer*. J Natl Cancer Inst, 2019. **111**(12): p. 1339-1349.
3. Knitz, M.W., et al., *Targeting resistance to radiation-immunotherapy in cold HNSCCs by modulating the Treg-dendritic cell axis*. J Immunother Cancer, 2021. **9**(4).
4. Bickett, T.E., et al., *FLT3L Release by Natural Killer Cells Enhances Response to Radioimmunotherapy in Preclinical Models of HNSCC*. Clin Cancer Res, 2021. **27**(22): p. 6235-6249.

5. Oweida, A., et al., *Resistance to Radiotherapy and PD-L1 Blockade Is Mediated by TIM-3 Upregulation and Regulatory T-Cell Infiltration*. Clin Cancer Res, 2018. **24**(21): p. 5368-5380.
6. Oweida, A., et al., *Ionizing radiation sensitizes tumors to PD-L1 immune checkpoint blockade in orthotopic murine head and neck squamous cell carcinoma*. Oncoimmunology, 2017. **6**(10): p. e1356153.
7. Reticker-Flynn, N.E., et al., *Lymph node colonization induces tumor-immune tolerance to promote distant metastasis*. Cell, 2022. **185**(11): p. 1924-1942 e23.
8. Cui, C., et al., *Neoantigen-driven B cell and CD4 T follicular helper cell collaboration promotes anti-tumor CD8 T cell responses*. Cell, 2021. **184**(25): p. 6101-6118 e13.
9. Guo, Y., et al., *Metabolic reprogramming of terminally exhausted CD8(+) T cells by IL-10 enhances anti-tumor immunity*. Nat Immunol, 2021. **22**(6): p. 746-756.
10. Liu, Y., et al., *IL-2 regulates tumor-reactive CD8(+) T cell exhaustion by activating the aryl hydrocarbon receptor*. Nat Immunol, 2021. **22**(3): p. 358-369.
11. Hui, C., et al., *Overcoming Resistance to Immunotherapy in Head and Neck Cancer Using Radiation: A Review*. Front Oncol, 2021. **11**: p. 592319.
12. Weiss, J., et al., *Concurrent Definitive Immunoradiotherapy for Patients with Stage III-IV Head and Neck Cancer and Cisplatin Contraindication*. Clin Cancer Res, 2020. **26**(16): p. 4260-4267.
13. Campian, J.L., et al., *Association between severe treatment-related lymphopenia and progression-free survival in patients with newly diagnosed squamous cell head and neck cancer*. Head Neck, 2014. **36**(12): p. 1747-53.
14. Dai, D., et al., *The impact of radiation induced lymphopenia in the prognosis of head and neck cancer: A systematic review and meta-analysis*. Radiother Oncol, 2022. **168**: p. 28-36.
15. Damen, P.J.J., et al., *The Influence of Severe Radiation-Induced Lymphopenia on Overall Survival in Solid Tumors: A Systematic Review and Meta-Analysis*. Int J Radiat Oncol Biol Phys, 2021. **111**(4): p. 936-948.
16. Klages, K., et al., *Selective depletion of Foxp3+ regulatory T cells improves effective therapeutic vaccination against established melanoma*. Cancer Res, 2010. **70**(20): p. 7788-99.
17. Dixon, M.L., et al., *Remodeling of the tumor microenvironment via disrupting Blimp1(+) effector Treg activity augments response to anti-PD-1 blockade*. Mol Cancer, 2021. **20**(1): p. 150.
18. Gkoutidi, A.O., et al., *MHC Class II Antigen Presentation by Lymphatic Endothelial Cells in Tumors Promotes Intratumoral Regulatory T cell-Suppressive Functions*. Cancer Immunol Res, 2021. **9**(7): p. 748-764.
19. Schaeue, D., et al., *Maximizing tumor immunity with fractionated radiation*. Int J Radiat Oncol Biol Phys, 2012. **83**(4): p. 1306-10.
20. Filatenkov, A., et al., *Treatment of 4T1 metastatic breast cancer with combined hypofractionated irradiation and autologous T-cell infusion*. Radiat Res, 2014. **182**(2): p. 163-9.
21. Zhang, F., et al., *Optimal combination treatment regimens of vaccine and radiotherapy augment tumor-bearing host immunity*. Commun Biol, 2021. **4**(1): p. 78.

22. Saddawi-Konefka, R., et al., *Lymphatic-Preserving Treatment Sequencing with Immune Checkpoint Inhibition Unleashes cDC1-Dependent Antitumor Immunity in HNSCC*. bioRxiv, 2022: p. 2022.02.01.478744.
23. Arce Vargas, F., et al., *Fc-Optimized Anti-CD25 Depletes Tumor-Infiltrating Regulatory T Cells and Synergizes with PD-1 Blockade to Eradicate Established Tumors*. *Immunity*, 2017. **46**(4): p. 577-586.
24. Solomon, I., et al., *CD25-Treg-depleting antibodies preserving IL-2 signaling on effector T cells enhance effector activation and antitumor immunity*. *Nat Cancer*, 2020. **1**(12): p. 1153-1166.
25. Christophersen, A., *Peptide-MHC class I and class II tetramers: From flow to mass cytometry*. *HLA*, 2020. **95**(3): p. 169-178.
26. Rao, G.K., et al., *T cell LFA-1-induced proinflammatory mRNA stabilization is mediated by the p38 pathway kinase MK2 in a process regulated by hnRNPs C, H1 and K*. *PLoS One*, 2018. **13**(7): p. e0201103.
27. Fei, T., et al., *Genome-wide CRISPR screen identifies HNRNPL as a prostate cancer dependency regulating RNA splicing*. *Proc Natl Acad Sci U S A*, 2017. **114**(26): p. E5207-E5215.
28. Hobbs, R.P., et al., *Keratin-dependent regulation of Aire and gene expression in skin tumor keratinocytes*. *Nat Genet*, 2015. **47**(8): p. 933-8.
29. Rahman, M.A., et al., *HnRNP L and hnRNP LL antagonistically modulate PTB-mediated splicing suppression of CHRNA1 pre-mRNA*. *Sci Rep*, 2013. **3**: p. 2931.
30. Platten, M., et al., *A vaccine targeting mutant IDH1 in newly diagnosed glioma*. *Nature*, 2021. **592**(7854): p. 463-468.
31. Kandula, S. and C. Abraham, *LFA-1 on CD4+ T cells is required for optimal antigen-dependent activation in vivo*. *J Immunol*, 2004. **173**(7): p. 4443-51.
32. Matsumoto, G., et al., *Cooperation between CD44 and LFA-1/CD11a adhesion receptors in lymphokine-activated killer cell cytotoxicity*. *J Immunol*, 1998. **160**(12): p. 5781-9.
33. Shao, W., et al., *CD44/CD70 blockade and anti-CD154/LFA-1 treatment synergistically suppress accelerated rejection and prolong cardiac allograft survival in mice*. *Scand J Immunol*, 2011. **74**(5): p. 430-7.
34. Walling, B.L. and M. Kim, *LFA-1 in T Cell Migration and Differentiation*. *Front Immunol*, 2018. **9**: p. 952.
35. Benechet, A.P., et al., *T cell-intrinsic S1PR1 regulates endogenous effector T-cell egress dynamics from lymph nodes during infection*. *Proc Natl Acad Sci U S A*, 2016. **113**(8): p. 2182-7.
36. Sahir, F., et al., *Development of a 43 color panel for the characterization of conventional and unconventional T-cell subsets, B cells, NK cells, monocytes, dendritic cells, and innate lymphoid cells using spectral flow cytometry*. *Cytometry A*, 2020.
37. Carozza, J.A., et al., *Extracellular cGAMP is a cancer cell-produced immunotransmitter involved in radiation-induced anti-cancer immunity*. *Nat Cancer*, 2020. **1**(2): p. 184-196.
38. Weisel, F.J., et al., *Germinal center B cells selectively oxidize fatty acids for energy while conducting minimal glycolysis*. *Nat Immunol*, 2020. **21**(3): p. 331-342.

39. Ferris, R.L., et al., *Neoadjuvant nivolumab for patients with resectable HPV-positive and HPV-negative squamous cell carcinomas of the head and neck in the CheckMate 358 trial*. *J Immunother Cancer*, 2021. **9**(6).
40. Wheat, W., et al., *Local immune and microbiological responses to mucosal administration of a Liposome-TLR agonist immunotherapeutic in dogs*. *BMC veterinary research*, 2019. **15**(1): p. 330.

REVIEWER COMMENTS

Reviewer #1 (Remarks to the Author):

The authors have addressed some of the experimental issues that I previously raised and performed a few new experiments. They also added statistical significance and number of mice used to the figures/results. However, statistical analysis is still poorly described in the method section and the input of a statistician is not apparent. In addition, the authors answer some questions by referring to other papers. For instance, although in Figure 1H there is no statistically significant difference between the groups, they state that “We reported that the p-value was 0.07 and showed a trend in tumor growth increase in the mice that had DLNs removed prior to therapy. The impact of this experiment is highlighted in a recent paper [22]”. While the fact that the findings of others (Ref 22 is a pre-publication and not a peer-reviewed study) may be supportive of the authors claim more than their own data, each manuscript should be evaluated on its own merits. The data in Figure 1H do not convincingly demonstrate what the authors claim and a “trend” is not sufficient. This experiment should be repeated – maybe with a larger number of mice powered to detect a real effect, or the data should be entirely removed. The authors also answer the request to show absolute numbers and not only percentages of immune cell subsets by stating that “Reporting percentages of populations is generally how immune data is presented in high impact journals”. This is not a valid argument: reporting a percentage may be the best choice in some circumstances, but this depends on the context and on the question asked. In this manuscript immune cell subsets are studied in irradiated nodes and in the blood of mice treated with large radiation fields – which are well-known to cause lymphopenia by killing the more radiation-sensitive cells – like naïve lymphocytes – therefore, in order to properly interpret the data in some of the experiments presented it is critical to know the number and not only the relative percentages of a specific cell subset. I agree that the inclusion of lymphoid tissue such as lymph nodes and spleen in the radiation field may be a contributor to the lymphopenia observed post-RT. However, the fact that the cellularity is unchanged in the irradiated DLN (new Suppl Fig 2C) is not consistent with this explanation, although it is not clear at which day post-radiation were the DLN assessed for cellularity, and whether the overall cellular composition is altered. The key question here is whether the peculiarity of the mouse model – which seems to have relatively larger radiation fields when irradiating draining nodes than it would be done in the clinic - produces results that are relevant to the clinical settings. I would like to point out that the blood flow velocity in the mouse carotid artery is 18 and 2.5 cm/s, in systole and diastole, respectively (PMID 19156686), thus in the 30 seconds that it takes to irradiate the nodes the amount of time for large blood volume exposure to RT is not minimal, as claimed by the authors. The human data added from a very interesting clinical study are nice but do not compensate for the flaws of the preclinical data. I would also suggest that the human data be placed not at the end but at the beginning of the manuscript since they show evidence of immune activation in the draining nodes of patients treated with tumor only RT and anti-PDL1. This does not do much to address the relationship between ENI and lymphopenia (given that no ENI is performed in the patients) but could be presented as the motivation for the studies performed in mice in this manuscript to test if ENI is detrimental. Additional issues remain, as detailed below.

1) It is now clear that the other models used, B16-OVA and 4T1, are irradiated without anti-CD25. The justification provided for this choice is unconvincing. Since the models are sensitive to RT+ICB (CTLA4 and/or PD1) it would have been more informative to test the response to these more clinically relevant immunotherapies. Most importantly, the question remains if the data are meaningful: B16-OVA lacks a control group without RT and there is no significant difference in distal tumor control between “Tumor only” and ENI (Figure 1). In 4T1 there is no difference in the primary tumor but there is a statistically significant difference in the distal tumor, a peculiar result considering that abscopal responses have not been reported in the literature with radiation alone in this model. Concerning the lung metastases only one representative image is shown in Supplementary Fig 1H, thus it is not possible to know if there are differences between “Tumor only” and ENI. Together with the fact that it is not stated if this experiment was repeated and the results reproducible, a lack of rigor still persists. In the text of results this experiment is presented as supporting the “generalizability” of the data with P029 tumor (Figure 1D) that “suggests that ENI accelerates metastatic growth” – quoting the authors.

This is a pretty consequential result – if true - that deserves to be supported by better data.

2) Figure 1I and evaluation of circulating immune cells: the entire panel of markers tested is now shown in Supplementary Fig 1J and K, but it is not mentioned or discussed in the text of results. The interpretation of the data should take into consideration also the relevant markers of T cell effector function that do not appear to be affected by ENI, such as IFN γ + CD4 and CD8 T cells, even if this is not consistent with the authors hypothesis.

3) Figure 2C: The strain of origin of LY2 cells (BALB/c mice) was not mentioned in Methods or text of results, so the explanation provided in the rebuttal is useful and should also be spelled out in the results section. That said, it is not clear if tumor control is affected by the adoptively transferred T cells. DO11 TCRtg CD4 T cells are lower (as % CD4) in blood of mice treated with ENI. This suggests that they accumulate in the tumor-draining nodes where they get killed by RT because this is where the antigen is presented, and Suppl Fig 5C supports this. However, without a control group of mice with a tumor that does not express OVA (or the analysis of non-tumor draining LN in mice with the OVA+ tumor) this interpretation remains somewhat speculative.

5) Figure 2E: decreased %CD8+ T cells among CD3+ T cells in blood is not very informative without knowing the total lymphocyte counts.

6) Figure 4: the priming of adoptively transferred OVA-specific CD4 T cells is not demonstrated to require the irradiation of the OVA-negative tumor. Because OVA-specific CD4 T cells are adoptively transferred an antigen encounter cannot be measured by clonal expansion. Thus, the reduction in OVA-specific CD4 T cells in blood (Figure 4C) of mice treated with ENI can be due to reduced survival of the transferred cells rather than lack of priming. A control group showing that the priming is reduced when the buccal tumor is not treated is required but not included. In addition, the most striking result is the almost total absence of CD3+ T cells in the flank tumor of the ENI-treated mice (panel G) while the differences measured in the inguinal nodes are much less pronounced. This is an interesting finding but it is not pursued. There is a description of the expression of CCR7 on CD4 T cells in the irradiated LN, but changes do not seem to be statistically significant between the two groups (Suppl Fig 5A) and the increased apoptosis in CD4 T cells and DCs in the irradiated LN is not surprising (Suppl Fig 5C and D) but the time point at which this is analyzed is not specified, and a kinetics analysis is required to understand the significance of this process.

7) As a minor point, on page 25 it is stated that “removal of the nodes after SBRT and anti-CD25 did not dampen the immune response (Figure 4N)” but Fig 4N does not exist, the correct figure is 3N.

8) Figure 3M: this is an important experiment with CD4 and CD8 depletion, but the information provided is insufficient to understand the data: how many mice were used per group? Where was the tumor implanted? How was depletion verified? Are these groups part of the experiment shown in Figure 3A?

9) The manuscript, including discussion, would benefit from the input of an immunologist – to put the role of CD4 T cells and other concepts in the right perspective.

Overall, while to their credit the authors have performed a lot of work in challenging mouse models that attempt to reproduce the clinic, and some of the data are convincing and important, the manuscript continues to suffer from issues of rigor and over/mis-interpretation of the results. The manuscript would greatly benefit from the removal of questionable data and unsupported claims and restructuring and focusing on the data showing that ENI reduces systemic responses.

Reviewer #2 (Remarks to the Author):

The authors have adequately responded to my previous concerns. I have no further comments.

Reviewer #3 (Remarks to the Author):

I feel that the authors have adequately addressed my concerns and those of the other reviewers. There are some remaining limitations, but these have been appropriately acknowledged. This is a tremendous amount of work and an important study.

Reviewer #4 (Remarks to the Author):

Dear Authors, thank you for addressing the specific points that were listed by the Reviewer #4.

Reviewer #1 (Remarks to the Author):

The authors have addressed some of the experimental issues that I previously raised and performed a few new experiments. They also added statistical significance and number of mice used to the figures/results. However, statistical analysis is still poorly described in the method section and the input of a statistician is not apparent.

We apologize for the lack of description. The additional requested statistical data have been added to the methods section. Dr. Zhuang, a biostatistician, has comprehensively reviewed all the statistics for this manuscript.

In addition, the authors answer some questions by referring to other papers. For instance, although in Figure 1H there is no statistically significant difference between the groups, they state that “We reported that the p-value was 0.07 and showed a trend in tumor growth increase in the mice that had DLNs removed prior to therapy. The impact of this experiment is highlighted in a recent paper [22]”. While the fact that the findings of others (Ref 22 is a pre-publication and not a peer-reviewed study) may be supportive of the authors claim more than their own data, each manuscript should be evaluated on its own merits. The data in Figure 1H do not convincingly demonstrate what the authors claim and a “trend” is not sufficient. This experiment should be repeated – maybe with a larger number of mice powered to detect a real effect, or the data should be entirely removed.

We appreciate your point as others have shown significance for a similar experimental design. The pre-print manuscript for reference 22 (now reference 14) is now published in this journal and the citation is included. Validation of the data by an independent group for any scientific question is a testament to the strength of the evidence. The data in Figure 1H show a trend towards significance consistent with the data presented in reference 22 and has been moved to Supplemental Figure 1J. This again has been reviewed by a statistician, Dr. Zhuang. We have also included a section in the Discussion section of the manuscript to address the limitations of this experiment.

The authors also answer the request to show absolute numbers and not only percentages of immune cell subsets by stating that “Reporting percentages of populations is generally how immune data is presented in high impact journals”. This is not a valid argument: reporting a percentage may be the best choice in some circumstances, but this depends on the context and on the question asked.

In this manuscript immune cell subsets are studied in irradiated nodes and in the blood of mice treated with large radiation fields – which are well-known to cause lymphopenia by killing the more radiation-sensitive cells – like naïve lymphocytes – therefore, in order to properly interpret the data in some of the experiments presented it is critical to know the number and not only the relative percentages of a specific cell subset.

Our apologies. We examined very closely numerous publications in this journal and the reference of high impact was simply to that. We have several well published, senior immunologists, on the author list of this manuscript, who agree with this standard of reporting. All collaborated on the reporting of the data. Further, reference number 22 was published in this journal, and we adhere by the same standard of reporting as that one or others. However, counts for tumor data are now included in Supplemental Figure 2D. Additionally, all our source data have been uploaded as raw files and available to the public for replicability.

I agree that the inclusion of lymphoid tissue such as lymph nodes and spleen in the radiation field may be a contributor to the lymphopenia observed post-RT. However, the fact that the cellularity is unchanged in the irradiated DLN (new Suppl Fig 2C) is not consistent with this explanation, although it is not clear at which day post-radiation were the DLN assessed for cellularity, and whether the overall cellular composition is altered.

The DLN histology samples were collected the same day as when the flow cytometry was done, and we have now made sure that that is explicitly stated in the figure legend. We evaluated the cellular composition in the DLNs (Figure 2F, Figure 2H). T cell activation was also assessed, which was markedly different. We agree that we would have expected a difference in cellularity between the two groups, but we have also now quantitated the size of the DLNs and again, we do not see a difference (Supplemental Figure 2C). This is likely due to the fact that the cells are still actively undergoing apoptosis due to the radiation treatment in the nodes, as we now show in Figure 4G.

The key question here is whether the peculiarity of the mouse model – which seems to have relatively larger radiation fields when irradiating draining nodes than it would be done in the clinic - produces results that are relevant to the clinical settings. I would like to point out that the blood flow velocity in the mouse carotid artery is 18 and 2.5 cm/s, in systole and diastole, respectively (PMID 19156686), thus in the 30 seconds that it takes to irradiate the nodes the amount of time for large blood volume exposure to RT is not minimal, as claimed by the authors. We appreciate the discussion on this topic as we often debate radiation fields for our various experiments. For these experiments, we believe that the data are relative. It is true that the field for ENI may be larger than in a human, but still surely contains <5% of the circulating blood at a given time. If circulation is fast, all of the blood will be there for <5% of the beam time and receive <5% of the dose, which is not enough to kill the lymphocytes. In contrast, there are large numbers of lymphocytes in the lymph nodes receiving the whole 5 Gy and likely source of lymphopenia.

Additionally, since the submission of this manuscript, another publication in Neoplasia (PMID 35667149) and directly addresses this question. Telarovic et al. (PMID 35667149) found that 1) large volume irradiation did produce prolonged lymphopenia in mice, 2) that this was due to tumor draining lymph node irradiation, not by the treatment volume and 3) that a smaller volume restricted to the TME increased lymphocyte recruitment to the TME. This has now been added to the Discussion section.

The human data added from a very interesting clinical study are nice but do not compensate for the flaws of the preclinical data. I would also suggest that the human data be placed not at the end but at the beginning of the manuscript since they show evidence of immune activation in the draining nodes of patients treated with tumor only RT and anti-PDL1. This does not do much to address the relationship between ENI and lymphopenia (given that no ENI is performed in the patients) but could be presented as the motivation for the studies performed in mice in this manuscript to test if ENI is detrimental.

Precisely because of what the reviewer cited in terms of relation between ENI and lymphopenia, the data were added at the end as clinical correlates. The head and neck cancer literature has ample documentation of treatment induced lymphopenia in the context of conventional

chemoradiation with ENI (e.g. PMID: 24174270; PMID: 35017020; PMID: 34329738; PMID: 34771432). This compared to none of the enclosed clinical trial at the same time points. We agree, however, that this is not a direct comparison and that would have to be done on trial. This is now integrated in the Discussion section.

Additional issues remain, as detailed below.

1) It is now clear that the other models used, B16-OVA and 4T1, are irradiated without anti-CD25. The justification provided for this choice is unconvincing. Since the models are sensitive to RT+ICB (CTLA4 and/or PD1) it would have been more informative to test the response to these more clinically relevant immunotherapies.

The point of this experiment was to show the effect of LN irradiation on the ability of radiation to activate systemic immune response, potentially offering an explanation for the “radioresistant” nature of these models. Telarovic et al. (2022) has demonstrated this in the context of immunotherapy. This has been added to Results section of the manuscript.

Most importantly, the question remains if the data are meaningful: B16-OVA lacks a control group without RT and there is no significant difference in distal tumor control between “Tumor only” and ENI (Figure 1). In 4T1 there is no difference in the primary tumor but there is a statistically significant difference in the distal tumor, a peculiar result considering that abscopal responses have not been reported in the literature with radiation alone in this model. Concerning the lung metastases only one representative image is shown in Supplementary Fig 1H, thus it is not possible to know if there are differences between “Tumor only” and ENI. Together with the fact that it is not stated if this experiment was repeated and the results reproducible, a lack of rigor still persists.

In these experiments we were testing how RT to the tumor alone differed from RT to the tumor and bilateral neck nodes. A non-treated group for the B16-OVA been previously published with 100% take rate. Although the distant tumor size was not significant, there was a trend towards a difference and the local tumor growth difference was repeated in the LY2-OVA model. This suggests that antigenicity could be driving response in the local tumor.

A direct comparison to other 4T1 irradiation would be challenging if not impossible as others likely had the inguinal node within their irradiation field while we specifically designed the fields with CT guided imaging and fluoroscopy to omit inguinal or axillary node irradiation. Histological quantification is now added to corroborate the radiographic findings (Supplemental Figure 1H). These points have been updated in the results section of the manuscript. Additionally, given that the focus of this manuscript is head and neck cancer, both the B16 and the 4T1 data have now been moved to Supplementary Figure 1F,1G.

In the text of results this experiment is presented as supporting the “generalizability” of the data with P029 tumor (Figure 1D) that “suggests that ENI accelerates metastatic growth” – quoting the authors. This is a pretty consequential result – if true - that deserves to be supported by better data.

These data were generated, and repeated, with much rigor. Agree that this carries significant consequences. National trials are currently ongoing to address this question in humans, especially in light of the recently published article in this journal (reference 22) and the newly

added reference Telarovic et al. (PMID 35667149).

2) Figure 1I and evaluation of circulating immune cells: the entire panel of markers tested is now shown in Supplementary Fig 1J and K, but it is not mentioned or discussed in the text of results. The interpretation of the data should take into consideration also the relevant markers of T cell effector function that do not appear to be affected by ENI, such as IFN γ + CD4 and CD8 T cells, even if this is not consistent with the authors hypothesis.

Detailed description has now been added.

3) Figure 2C: The strain of origin of LY2 cells (BALB/c mice) was not mentioned in Methods or text of results, so the explanation provided in the rebuttal is useful and should also be spelled out in the results section. That said, it is not clear if tumor control is affected by the adoptively transferred T cells. DO11 TCRtg CD4 T cells are lower (as % CD4) in blood of mice treated with ENI. This suggests that they accumulate in the tumor-draining nodes where they get killed by RT because this is where the antigen is presented, and Suppl Fig 5C supports this. However, without a control group of mice with a tumor that does not express OVA (or the analysis of non-tumor draining LN in mice with the OVA+ tumor) this interpretation remains somewhat speculative.

Additional methods for the LY2 strain have been added to the methods section. We have now included analysis of cell death in the inguinal nodes which were not treated with radiation as a control (Supplemental Figure 5D).

5) Figure 2E: decreased %CD8+ T cells among CD3+ T cells in blood is not very informative without knowing the total lymphocyte counts.

We have added this as a limitation to our studies in the discussion.

6) Figure 4: the priming of adoptively transferred OVA-specific CD4 T cells is not demonstrated to require the irradiation of the OVA-negative tumor. Because OVA-specific CD4 T cells are adoptively transferred an antigen encounter cannot be measured by clonal expansion. Thus, the reduction in OVA-specific CD4 T cells in blood (Figure 4C) of mice treated with ENI can be due to reduced survival of the transferred cells rather than lack of priming. A control group showing that the priming is reduced when the buccal tumor is not treated is required but not included.

We have acknowledged this as a limitation of the study.

In addition, the most striking result is the almost total absence of CD3+ T cells in the flank tumor of the ENI-treated mice (panel G) while the differences measured in the inguinal nodes are much less pronounced. This is an interesting finding but it is not pursued.

We agree with the reviewer that that is where the impact lies. We pursued this by showing decreases in antigen specific T cells and activation in the nodes. We also examined if these T cells are antigen specific or not in the flank and found that there were less antigen specific T cells in the node as well. Using tSNE visualization we also show that the T cells that are in the flank tumor of the ENI treated mice are not active.

There is a description of the expression of CCR7 on CD4 T cells in the irradiated LN, but changes do not seem to be statistically significant between the two groups (Suppl Fig 5A) and the increased apoptosis in CD4 T cells and DCs in the irradiated LN is not surprising (Suppl Fig

5C and D) but the time point at which this is analyzed is not specified, and a kinetics analysis is required to understand the significance of this process.

Correct. The changes in CCR7 and CD4 T cells were not significant. The histology was performed on samples collected on the same day that the flow cytometry was run. We have added the lack of kinetics as a limitation of this analysis.

7) As a minor point, on page 25 it is stated that “removal of the nodes after SBRT and anti-CD25 did not dampen the immune response (Figure 4N)” but Fig 4N does not exist, the correct figure is 3N.

This has been corrected.

8) Figure 3M: this is an important experiment with CD4 and CD8 depletion, but the information provided is insufficient to understand the data: how many mice were used per group? Where was the tumor implanted? How was depletion verified? Are these groups part of the experiment shown in Figure 3A?

We apologize for the oversight. Numbers of mice are now included in the legend. The tumor was implanted in the left buccal. Depletion was verified using flow. This has all been included in the legend.

9) The manuscript, including discussion, would benefit from the input of an immunologist – to put the role of Cd4 T cells and other concepts in the right perspective.

Additional discussion has been added as requested.

Overall, while to their credit the authors have performed a lot of work in challenging mouse models that attempt to reproduce the clinic, and some of the data are convincing and important, the manuscript continues to suffer from issues of rigor and over/mis-interpretation of the results. The manuscript would greatly benefit from the removal of questionable data and unsupported claims and restructuring and focusing on the data showing that ENI reduces systemic responses.

Thank you. Every attempt has been made to do just that.

Reviewer #2 (Remarks to the Author):

The authors have adequately responded to my previous concerns. I have no further comments.

Reviewer #3 (Remarks to the Author):

I feel that the authors have adequately addressed my concerns and those of the other reviewers. There are some remaining limitations, but these have been appropriately acknowledged. This is a tremendous amount of work and an important study.

Reviewer #4 (Remarks to the Author):

Dear Authors, thank you for addressing the specific points that were listed by the Reviewer #4.

REVIEWERS' COMMENTS

Reviewer #1 (Remarks to the Author):

The issues that I raised and were meant to improve the manuscript have not been taken seriously by the authors.

Reviewer #3 (Remarks to the Author):

The authors have addressed all concerns appropriately.